

# Importance of Dry Deposition Parameterization Choice in Global Simulations of Surface Ozone

Anthony Y.H. Wong[1], Jeffrey A. Geddes[1], Amos P.K. Tai[2,3], Sam J. Silva[4]

[1]Department of Earth and Environment, Boston University, Boston, MA, USA
[2]Earth System Science Programme, Faculty of Science, The Chinese University of Hong Kong, Hong Kong
[3]Institute of Energy, Environment and Sustainability, and State Key Laboratory of Agrobiotechnology, The Chinese University of Hong Kong, Hong Kong
[4]Department of Civil and Environmental Engineering, Massachusetts Institute of Technology, Cambridge, MA, USA

*Correspondence to*: Jeffrey A. Geddes (jgeddes@bu.edu)

**Abstract.** Dry deposition is the second largest sink of tropospheric ozone. Increasing evidence has shown that ozone dry deposition actively links meteorology and hydrology with ozone air quality. However, there is little systematic investigation on the performance of different ozone dry deposition parameterizations at the global scale, and how parameterization choice can impact surface ozone simulations. Here we present the results of the first global, multi-decade modelling and evaluation of ozone dry deposition velocity ($v_d$) using multiple ozone dry deposition parameterizations. We use consistent assimilated meteorology and satellite-derived leaf area index (LAI) to simulate $v_d$ over 1982-2011 driven by four sets of ozone dry deposition parametrization that are representative of the current approaches of global ozone dry deposition modelling, such that the differences in simulated $v_d$ are entirely due to differences in deposition model structures. In addition, we use the surface ozone sensitivity to $v_d$ predicted by a chemical transport model to estimate the impact of mean and variability of ozone dry deposition velocity on surface ozone. Our estimated $v_d$ from four different parameterizations are evaluated against field observations, and while performance varies considerably by land cover types, our results suggest that none of the parameterizations are universally better than the others. Discrepancy in simulated mean $v_d$ among the parameterizations is estimated to cause 2 to 5 ppbv of discrepancy in surface ozone in the Northern Hemisphere (NH) and up to 8 ppbv in tropical rainforest in July, and up to 8 ppbv in tropical rainforests and seasonally dry tropical forests in Indochina in December. Parameterization-specific biases based on individual land cover type and hydroclimate are found to be the two main drivers of such discrepancies. We find statistically significant trends in the multiannual time series of simulated July daytime $v_d$ in all parameterizations, driven by warming and drying (southern Amazonia, southern African savannah and Mongolia) or greening (high latitudes). The trends in July daytime $v_d$ is estimated to be 1 % yr⁻¹ and leads to up to 3 ppbv of surface ozone changes over 1982-2011. The interannual coefficient of variation (CV) of July daytime mean $v_d$ in NH is found to be 5%-15%, with spatial distribution that varies with the dry deposition parameterization. Our sensitivity simulations suggest this can contribute between 0.5 to 2 ppbv to interannual variability (IAV) in surface ozone, but all models tend to underestimate interannual CV when compared to long-term ozone flux observations. We also find that IAV in some dry deposition parameterizations are more sensitive to LAI while others are more sensitive to climate. Comparisons with other published estimates of the IAV of



background ozone confirm that ozone dry deposition can be an important part of natural surface ozone variability. Our results
demonstrate the importance of ozone dry deposition parameterization choice on surface ozone modelling, and the impact of
IAV of $v_d$ on surface ozone, thus making a strong case for further measurement, evaluation and model-data integration of
ozone dry deposition on different spatiotemporal scales.
**1 Introduction**
Surface ozone ($O_3$) is one of the major air pollutants that poses serious threats to human health (Jerrett et al., 2009) and plant
productivity (Ainsworth et al., 2012; Reich, 1987; Wittig et al., 2007). Ozone exerts additional pressure on global food security
and public health by damaging agricultural ecosystems and reducing crop yields  (Avnery et al., 2011; McGrath et al., 2015;
Tai et al., 2014). Dry deposition, by which atmospheric constituents are removed from the atmosphere and transferred to the
Earth's surface through turbulent transport or gravitational settling, is the second-largest and terminal sink of tropospheric $O_3$
(Wild, 2007). Terrestrial ecosystems are particularly efficient at removing $O_3$ via dry deposition through stomatal uptake and
other non-stomatal pathways (Wesely and Hicks, 2000) (e.g., cuticle, soil, reaction with biogenic volatile organic compounds
(BVOCs) (Fares et al., 2010; Wolfe et al., 2011). Meanwhile, stomatal uptake of $O_3$ inflicts damage on plants by initiating
reactions  that impair their photosynthetic and stomatal regulatory capacity (Hoshika et al., 2014; Lombardozzi et al., 2012;
Reich, 1987). Widespread plant damage has the potential to alter the global water cycle (Lombardozzi et al., 2015) and suppress
the land carbon sink (Sitch et al., 2007), as well as to generate a cascade of feedbacks that affect atmospheric composition
including ozone itself (Sadiq et al., 2017; Zhou et al., 2018). Ozone dry deposition is therefore key in understanding how
meteorology (Kavassalis and Murphy, 2017), climate, and land cover change (Fu and Tai, 2015; Ganzeveld et al., 2010; Geddes
et al., 2016; Heald and Geddes, 2016; Sadiq et al., 2017; Sanderson et al., 2007; Young et al., 2013) can affect air quality and
atmospheric chemistry at large.

Analogous to other surface-atmosphere exchange processes (e.g., sensible and latent heat flux), $O_3$ dry deposition flux ($F_{O3}$)
is often expressed as the product of ambient $O_3$ concentrations at the surface ($[O_3]$) and a transfer coefficient (dry deposition
velocity, $v_d$) that describes the efficiency of transport (and removal) to the surface from the measurement height:
$$F_{O_3} = -[O_3]v_d \quad (1)$$
Also analogous to other surface fluxes, $F_{O3}$, $[O_3]$, and hence $v_d$ can be directly measured by the eddy covariance (EC) method
(e.g. Fares et al., 2014; Gerosa et al., 2005; Lamaud et al., 2002; Munger et al., 1996; Rannik et al., 2012) with random
uncertainty of about 20% (Keronen et al., 2003; Muller et al., 2010). Apart from EC, $F_{O3}$ and $v_d$ can also be estimated from
the vertical profile of $O_3$ by exploiting flux-gradient relationship (Foken, 2006) (termed the gradient method, GM) (e.g. Gerosa
et al., 2017; Wu et al., 2016, 2015). A recent review (Silva and Heald, 2018) has complied 75 sets of ozone deposition
measurement from the EC and GM methods across different seasons and land cover types over the past 30 years.





At the site level, ozone dry deposition over various terrestrial ecosystems can be simulated comprehensively by 1-D chemical
transport models (Ashworth et al., 2015; Wolfe et al., 2011; Zhou et al., 2017), which are able to account for vertical gradients
inside the canopy environment, and gas-phase reaction with BVOCs in addition to surface sinks. Regional and global models,
which lack the fine-scale information (e.g. vertical structure of canopy, in-canopy BVOCs emissions) and horizontal resolution
for resolving the plant canopy in such detail, instead rely on parameterizing $v_d$ as a network of resistances, which account for
the effects of turbulent mixing via aerodynamic ($R_a$), molecular diffusion via quasi-laminar sublayer resistances ($R_b$), and
surface sinks via surface resistance ($R_c$):
$$v_d = \frac{1}{R_a + R_b + R_c} \quad (2)$$

A diverse set of parameterizations of ozone dry deposition are available and used in different models and monitoring networks.
Examples include the Wesely parameterization (1989) and modified versions of it (e.g. Wang et al., 1998), the Zhang et al.
parameterization (Zhang et al., 2003), the Deposition of $O_3$ for Stomatal Exchange model (Emberson et al., 2000; Simpson et
al., 2012), and the Clean Air Status and Trends Network (CASTNET) deposition estimates (Meyers et al., 1998). The
calculation of $R_a$ and $R_b$ across these parameterizations often follow a standard formulation from micrometeorology (Foken,
2006; Wesely and Hicks, 1977, 2000; Wu et al., 2011) and thus does not vary significantly. The main difference between the
ozone dry deposition parameterizations lies on the surface resistance $R_c$. This resistance includes stomatal resistance ($R_s$),
which can be computed by a Jarvis-type multiplicative algorithm (Jarvis, 1976) where $R_s$ is the product of its minimum value
and a series of response functions to individual environmental conditions. Such conditions typically include air temperature
($T$), photosynthetically available radiation ($PAR$), vapour pressure deficit ($VPD$) and soil moisture ($\theta$), with varying complexity
and functional forms.

An advance of these efforts includes harmonizing $R_s$ with that computed by land surface models (Ran et al., 2017a; Val Martin
et al., 2014), which calculate $R_s$ by coupled photosynthesis-stomatal conductance ($A_n$-$g_s$) models (Ball et al., 1987; Collatz et
al., 1992, 1991). Such coupling should theoretically give a more realistic account of ecophysiological controls on $R_s$. Indeed,
it has been shown that the above approach may better simulate $v_d$ than the multiplicative algorithms that only considers the
effects $T$ and $PAR$ (Val Martin et al., 2014; Wu et al., 2011). The non-stomatal part of $R_c$ often consists of cuticular ($R_{cut}$),
ground ($R_g$) and other miscellaneous types of resistances (e.g., lower canopy resistance ($R_{lc}$) in Wesely (1989)). Due to very
limited measurements and mechanistic understanding towards non-stomatal deposition, non-stomatal resistances are often
constants (e.g., $R_g$) or simply scaled with leaf area index (LAI) (e.g., $R_{cut}$) (Simpson et al., 2012; Wang et al., 1998; Wesely,
1989), while some of the parameterizations (Zhang et al., 2003; Zhou et al., 2017) incorporate the observation of enhanced
cuticular $O_3$ uptake under leaf surface wetness (Altimir et al., 2006; Potier et al., 2015, 2017; Sun et al., 2016).



Various efforts have been made to evaluate and assess the uncertainty in modelling ozone dry deposition using field
measurements. Hardacre et al. (2015) evaluate the performance of simulated monthly mean $v_d$ and $F_{O3}$ by 15 chemical transport
models (CTM) from the Task Force on Hemispheric Transport of Air Pollutant (TF HTAP) against seven long-term site
measurements, 15 short-term site measurements, and modelled $v_d$ from 96 CASTNET sites. This work found that the seasonal
cycle is well-simulated across models, while demonstrating that the difference in land cover classification is the main source
of discrepancy between models. In this case, most of the models in TF HTAP use the same class of dry deposition
parameterization (Wang et al., 1998; Wesely, 1989), so a global evaluation of *different* deposition parameterizations was not
possible. Also, the focus in this intercomparison study was on seasonal, but not other (e.g. diurnal, daily, interannual)
timescales. Using an extended set of measurements, Silva and Heald (2018) evaluate the $v_d$ output from the Wang et al. (1998)
parameterization used by the GEOS-Chem chemical transport model. They show that diurnal and seasonal cycles are generally
well-captured, while the daily variability is not well-simulated. They find that differences in land type and LAI, rather than
meteorology, are the main reason behind model-observation discrepancy at the seasonal scale, and eliminating this model bias
results in up to 15% change in surface $O_3$. This study is also limited to a single parameterization. Using parameterizations that
are explicitly sensitive to other environmental variables (e.g. Simpson et al., 2012; Zhang et al., 2003) could conceivably lead
to different conclusions.

Other efforts have been made to compare the performance of different parameterizations. Centoni (2017) find that two different
dry deposition parameterizations, Wesely (1989) versus Zhang et al. (2003), implemented in the same chemistry-aerosol model
(United Kingdom Chemistry Aerosol model, UKMA), result in up to a 20% difference in simulated surface $O_3$ concentration.
This study demonstrates that uncertainty in $v_d$ can have large potential effect on surface $O_3$ simulation. Wu et al. (2018)
compare $v_d$ simulated by five North-American dry deposition parametrizations to a long-term observational record at a single
mixed forest in southern Canada, and find a large spread between the simulated $v_d$, with no single parameterization uniformly
outperforming others. They further acknowledge that as each parameterization is developed with its own set of limited
observations, it is natural that their performance can vary considerably under different environments, and advocate for an
"ensemble" approach to dry deposition modelling. This highlights the importance of parameterization choice as a key source
of uncertainty in modelling ozone dry deposition. Meanwhile, in another evaluation at a single site, Clifton et al. (2017) show
that the GEOS-Chem parameterization largely underestimates the interannual variability (IAV) of $v_d$ in Harvard Forest based
on the measurement from 1990 to 2000, although they were unable to conclude how the IAV of $v_d$ may contribute to the IAV
of $O_3$.

These developments have made a substantial contribution to our understanding of the importance of $O_3$ dry deposition in
atmospheric chemistry models. Still, pertinent questions remain about the impact of dry deposition model physics on
simulations of the global distribution of ozone and its long-term variability. Here, we build on previous works by posing and
answering the following questions:





1)  How does the global distribution of mean $v_d$ vary with different dry deposition parameterizations, and what drives the discrepancies among them? How much might the choice of deposition parameterization affect spatial distribution of surface ozone concentration simulated by a chemical transport model?

2)  How are the IAV and long-term trends of $v_d$ different across deposition parameterizations, and what drives the discrepancies among them? Do they potentially contribute different predictions of the long-term temporal variability in surface ozone?

The answers to such question could have important consequences on our ability to predict long-term changes in atmospheric $O_3$ concentrations as a function of changing climate and land cover characteristics. In general, there is a high computational cost to thorough and large-scale evaluations of different dry deposition parameterizations embedded in CTMs. In this study, we explore these questions using a strategy that combines an offline dry deposition modelling framework incorporating long-term assimilated meteorological and land surface remote sensing data, in combination with a set of CTM sensitivity simulations.

## 2 Method

### 2.1 Dry deposition parameterization

A detailed description of the common dry deposition parameterizations we explore can be found in Wu et al. (2018). Here we consider several "big-leaf" models commonly used by global chemical transport models. More complex multilayer models require the vertical profiles of leaf area density for different biomes which are generally not available for regional and global models. From the wide range of literature on dry deposition studies, we observe that $R_s$ is commonly modelled through one of the following approaches:

1)  Multiplicative algorithm that considers the effects of LAI, temperature and radiation (Wang et al., 1998).

2)  Multiplicative algorithm that considers the effects of LAI, temperature, radiation and water stress (e.g. Meyers et al., 1998; Pleim and Ran, 2011; Simpson et al., 2012; Zhang et al., 2003).

3)  Coupled $A_n$-$g_s$ model, which exploit the strong empirical relationship between photosynthesis ($A_n$) and stomatal conductance ($g_s$) and to simulate $A_n$ and $g_s = 1/R_s$ simultaneously (e.g. Ran et al., 2017b; Val Martin et al., 2014).

Similarly, their functional dependence of non-stomatal surface resistances can be classified into two classes:

1)  Mainly scaling with LAI, with in-canopy aerodynamics parameterized as function of friction velocity ($u_*$) or radiation (Meyers et al., 1998; Simpson et al., 2012; Wang et al., 1998)

2)  Additional dependence of cuticular resistance on relative humidity (Pleim and Ran, 2011; Zhang et al., 2003)

With these considerations, we identify four common parameterizations that are representative of the types of approaches described above:



1) The version of Wesely (1989) with the modification from Wang et al. (1998) (hereafter referred to as W98), which is used extensively in global CTMs (Hardacre et al., 2015) and comprehensively discussed by Silva and Heald (2018). This represents Type 1 in both stomatal and non-stomatal parametrizations.

2) The Zhang et al. (2003) parameterization (hereafter referred to as Z03), which is used in many North American air quality modelling studies (e.g. Huang et al., 2016; Kharol et al., 2018) and Canadian Air and Precipitation Monitoring Network (CAPMoN) (e.g. Zhang et al., 2009). This represents Type 2 in both stomatal and non-stomatal parameterizations

3) W89 with $R_s$ calculated from a widely-used coupled $A_n$-$g_s$ model, the Ball-Berry model (hereafter referred to as W98_BB) (Ball et al., 1987; Collatz et al., 1992, 1991), which is similar to that proposed by Val Martin et al. (2014). This represents Type 3 in stomatal and Type 1 in non-stomatal parametrization.

4) Z03 with the Ball-Berry model (Z03_BB), which is comparable to the configuration in Centoni (2017). This represents Type 3 in stomatal and Type 2 in non-stomatal parametrization.

Another important consideration in choosing Z03 and W98 is that they both have open-source parameters for all major land types over the globe, making them widely applicable in global modelling. We extract the source code (Wang et al., 1998) and parameters (Baldocchi et al., 1987; Jacob et al., 1992; Jacob and Wofsy, 1990; Wesely, 1989) of W98 from GEOS-Chem CTM (http://wiki.seas.harvard.edu/geos-chem/index.php/Dry_deposition). The source code of Z03 are obtained through personal communication with Zhiyong Wu and Leiming Zhang, which follows the series of papers that described the development and formalism of the parameterization (Brook et al., 1999; Zhang et al., 2001, 2002, 2003). The Ball-Berry $A_n$-$g_s$ model (Ball et al., 1987; Collatz et al., 1992, 1991; Farquhar et al., 1980) and its solver are largely based on the algorithm of CLM (Community Land Model) version 4.5 (Oleson et al., 2013), which is numerically stable (Sun et al., 2012). Since $R_c$ typically dominates the deposition velocity of $O_3$ (Fares et al., 2010; Wu et al., 2018), we use identical formulae of $R_a$ and $R_b$ (Paulson, 1970; Wesely and Hicks, 1977) for each individual parameterizations, allowing us to focus our analysis on differences in parameterizations of $R_c$ alone. Table A1 gives a brief description on the formalism of each of the dry deposition parameterizations.

**2.2 Dry deposition model configuration, inputs, and simulation**

The above parameterizations are re-implemented in R language (R core team, 2017) in the modeling framework of the Terrestrial Ecosystem Model in R (http://www.cuhk.edu.hk/sci/essc/tgabi/tools.html), and driven by gridded surface meteorology and land surface data sets. The meteorological forcing chosen for this study is the Modern-Era Retrospective Analysis for Research and Application-2 (MERRA-2) (Gelaro et al., 2017), an assimilated meteorological product at hourly time resolution spanning from 1980 to present day. MERRA-2 contains all the required surface meteorological fields except *VPD* and *RH*, which can be readily computed from *T,* specific humidity (*q*) and surface air pressure (*P*). We use the CLM land surface dataset (Lawrence and Chase, 2007), which contains information for land cover, per-grid cell coverage of each plant





functional type (PFT), PFT-specific LAI and soil property. CLM land types are mapped to the land type of W98 following
Geddes et al. (2016). The mapping between CLM and Z03 land types are given in Table A2. Other relevant vegetation and
soil parameters (e.g. leaf physiological and soil hydraulic constants) are also imported from CLM 4.5 (Oleson et al., 2013),
while land cover specific $z_0$ values follow Geddes et al. (2016).

As the IAV of LAI could be an important factor in simulating $v_d$, the widely-used third generation Global Inventory Modelling
and Mapping Studies Leaf Area Index product (GIMMS LAI3g, abbreviated as LAI3g in this paper) (Zhu et al., 2013), which
is a global time series of LAI with 15-day temporal frequency and 1/12 degree spatial resolution spanning from late 1981 to
2011, is incorporated in this study. We use this data set to derive interannual scaling factors that can be applied to the baseline
CLM-derived LAI (Lawrence and Chase, 2007). All the input data are aggregated into horizontal resolution of 2°×2.5° to align
with the CTM sensitivity simulation described in next sub-section. To represent sub-grid land cover heterogeneity, grid cell-
level $v_d$ is calculated as the sum of $v_d$ over all sub-grid land types weighted by their percentage coverage in the grid cell (a.k.a
tiling or mosaic approach, e.g. Li et al., 2013). This reduces the information loss when land surface data is aggregated to
coarser spatial resolution, and allows us to retain PFT-specific results for each grid box in the offline dry deposition
simulations.

We run three sets of 30-years (1982-2011) simulations with the deposition parameterizations to investigate the how $v_d$
simulated by different parameterizations responds to different environmental factors over multiple decades. The settings of the
simulations are summarized in Table 1. The first set, [Clim], focuses on meteorological variability alone, driven by MERRA-
2 meteorology and a multiyear (constant) mean annual cycle of LAI derived from LAI3g. The second set, [Clim+LAI],
combines the effects of meteorology and IAV in LAI, driven by the same MERRA-2 meteorology plus the LAI time series
from LAI3g. As the increase atmospheric $CO_2$ level over multidecadal timescales may lead to significant reduction in $g_s$ as
plants tend to conserve water (e.g. Franks et al., 2013; Rigden and Salvucci, 2017), we introduce the third set of simulation,
[Clim+LAI+$CO_2$], which is driven by varying meteorology and LAI, plus the annual mean atmospheric $CO_2$ level measured
in Mauna Loa (Keeling et al., 2001) (for the first two sets of simulations, atmospheric $CO_2$ concentration held constant at 390
ppm). Since W98 and Z03 do not respond to changes in $CO_2$ level, only W98_BB and Z03_BB are run with [Clim+LAI+$CO_2$]
to evaluate this impact. We focus on the daytime (solar elevation angle > 20°) $v_d$, as both $v_d$ and surface $O_3$ concentration
typically peak around this time. We calculate monthly means, filtering out the grid cells with monthly total daytime < 100
hours, which would be an indication of dormant biosphere.

In summary, we present for the first time a unique set of global dry deposition velocity predictions over the last 30 years driven
by identical meteorology and land cover, so that discrepancies (in space and time) among the predicted $v_d$ are a result
specifically of dry deposition parameterizations alone.



### 2.3 Chemical transport model sensitivity experiments

We quantify the sensitivity of surface $O_3$ to variations in $v_d$ using a global 3D CTM, GEOS-Chem version 11.01 (Bey et al., 2001), which includes comprehensive $HO_x$-$NO_x$-VOC-$O_3$-$BrO_x$ chemical mechanisms (Mao et al., 2013) and is widely used to study tropospheric ozone (e.g. Hu et al., 2017; Travis et al., 2016; Zhang et al., 2010). The model is driven by the assimilated meteorological data from the GEOS-FP (Forward Processing) Atmospheric Data Assimilation System (GEOS-5 ADAS) (Rienecker et al., 2008), which is jointly developed by National Centers for Environmental Prediction (NCEP) and the Global Modelling and Assimilation Office (GMAO). The model is run with a horizontal resolution of 2°×2.5°, and 47 vertical layers. The dry deposition module, which has been discussed above (W98), is driven by the monthly mean LAI retrieved from Moderate Resolution Imaging Spectroradiometer (MODIS) (Myneni et al., 2002) and the 2001 version of Olson land cover map (Olson et al., 2001). Both of the maps are binned from their native resolutions to 0.25°×0.25°.

We propose to estimate the sensitivity of surface $O_3$ concentrations to uncertainty/changes in $v_d$ by the following equation:

$$\Delta O_3 = \beta \frac{\Delta v_d}{v_d}$$

where $\Delta O_3$ is the response of monthly mean daytime surface $O_3$ to fractional change in $v_d$ ($\Delta v_d/v_d$), and $\beta$ accounts for the sensitivity of surface $O_3$ concentration in a grid box to the perturbation in $v_d$ within that grid box. To estimate $\beta$, we run two simulations for the year 2013, one with default setting and another where we perturb $v_d$ by +30%. Since not every gaseous species deposit with the same functional relationships as $O_3$, we only adjust the $v_d$ of $O_3$ to avoid perturbing the chemistry resulting from the deposition of other chemically relevant species (e.g. PAN, $HNO_3$). Thus, this approach could represent a conservative estimate of $O_3$ sensitivity to $v_d$ if the impacts on other species result in additional effects on $O_3$. Nevertheless, we use this sensitivity to estimate the potential impact of $v_d$ simulation on surface $O_3$ concentration to a first order in subsequent sections. This approach is based on the reasonably linear response of surface $O_3$ to $v_d$ over comparable range of $v_d$ change (Wong et al., 2018). We limit our analysis to grid cells where the monthly average $v_d$ is greater than 0.25 cm s$^{-1}$ in the baseline simulation, since changes in surface $O_3$ elsewhere are expected to be attributed more to chemical transport rather than the local perturbation of $v_d$ (Wong et al., 2018).

### 3. Evaluation of Dry Deposition Parameterizations

We first compare our offline simulations of seasonal mean daytime average $v_d$ that result from the four parameterizations in the [Clim] and [Clim+LAI] scenarios with an observational database largely based on the evaluation presented in Silva and Heald (2018). We use two unbiased and symmetrical statistical metrics, normalized mean bias factor (*NMBF*) and normalized mean absolute error factor (*NMAEF*), to evaluate our parameterizations. Positive *NMBF* indicates that the parameterization overestimates the observations by a factor of 1 + *NMBF* and the absolute gross error is *NMAEF* times the mean observation, while negative *NMBF* implies that the parameterization underestimates the observations by a factor of 1 - *NMBF* and the





absolute gross error is *NMAEF* times the mean model prediction (Yu et al., 2006). We use the simulated subgrid land type-
specific predictions of $v_d$ that correctly match the land type and the averaging window indicated by the observations. We
exclude instances where the observed land type does not have a match within the model grid box. While this leads to a reduction
of dataset size comparing to Silva and Heald (2018), this means that mismatched land-cover types can be ignored as a factor
in model bias.

Figure 1 shows the fractional coverage within each grid cell and the geographic locations of $O_3$ flux observation sites for each
major land type. Nearly all the observations are clustered in Europe and North America, except three sites in the tropical
rainforest and one site in tropical deciduous forest in Thailand. The resulting *NMBF* and *NMAEF* for five major land type
categories are shown in Table 2, and the list of sites and their descriptions are given in Table A3. In general, the numerical
ranges of both *NMBF* and *NMAEF* are similar to that of Silva and Heald (2018), and no single parameterization of the four
parameterizations outperforms the others across all five major land types. Here, we focus on describing how our
implementation of the dry deposition parameterizations produce consistent comparisons with earlier results.

As summarized in Table 2, each parameterization shows distinct biases over specific land types (we subsequently refer to
this as the "land-type specific bias" unique to each parameterization). Comparing the two multiplicative parameterizations
(W98 and Z03), we find that W98 performs satisfactorily over deciduous forests and tropical rainforests, while strongly
underestimating daytime $v_d$ over coniferous forests. In contrast, Z03 performs better in coniferous forests but worse in
tropical rainforests and deciduous forests. The severe underestimation of daytime $v_d$ by Z03 over tropical rainforests has
previously been attributed to persistent canopy wetness, and hence stomatal blocking imposed by the parameterization
(Centoni, 2017). The simple linear *VPD* response function in Z03 may overestimate the sensitivity of $g_s$ to *VPD* under the
high temperature in tropical rainforest. We also note that even for the same location, $v_d$ can vary significantly between
seasons (Rummel et al., 2007) and management practices (Fowler et al., 2011), which models may fail to capture due to
limited representations of land cover. Given the small sample size (N = 5), diverse environments, and large anthropogenic
intervention in the tropics, the disparity in performance metrics may not fully reflect the relative model performance.
Baseline cuticular resistances in Z03 under dry and wet canopy are 1.5 and 2 times that of coniferous forests, respectively
(Zhang et al., 2003), such that the enhancement of cuticular uptake by wetness may not compensate the reduced $g_s$ over
tropical rainforests, and, to a lesser extent, deciduous forests. The higher cuticular uptake may explain the better performance
of Z03 over W98 over coniferous forests, where strong non-stomatal (though not necessarily cuticular) ozone sinks are often
observed (e.g. Gerosa et al., 2005; Wolfe et al., 2011).

Over grasslands, W98 has higher positive biases, while Z03 has higher absolute errors. This is because for datasets at high
latitudes, the dominant grass PFT is arctic grass, which is mapped to "tundra" land type (Geddes et al., 2016). While tundra
is parameterized similarly to grasslands in W98, this is not the case in Z03. Combined with the general high biases at other





sites for these parameterizations, the large low biases for "tundra" sites in Z03 lower the overall high biases but leads to
higher absolute errors.

Over croplands, the positive biases and absolute errors are relatively large for both W98 and Z03 (with Z03 performing
worse in general than W98). This may be attributed to the lack of response to *VPD* over all crop and grass land types in Z03.
The functional and physiological diversity with the "crop" land type also contributes to the general difficulty in simulating $v_d$
over cropland. Even though Z03 has individual parameterizations for 4 specific crop types (rice, sugar, maize and cotton),
this advantage is difficult to fully leverage as most global land cover data sets do not resolve croplands into such detail.

Substituting the native $g_s$ in W98 and Z03 by that simulated by Ball-Berry model (the W98_BB and Z03_BB runs)
generally, though not universally, leads to improvement in model performance against the observations. W98_BB has
considerably smaller biases and absolute errors than W98 over grassland. While having little effect on the absolute error,
W98_BB improves the biases over coniferous forest and cropland compared to W98, but worsens the biases over rainforests
and deciduous forests. In contrast, Z03_BB is able to improve the model-observation agreement over all 5 land types when
compared to Z03. This finding echoes that from Wu et al. (2011), who explicitly show the advantage of replacing the $g_s$ of
Wesely (1989) with the Ball-Berry model in simulating $v_d$ over a forest site, and in addition shows the potential of Ball-Berry
model in improving spatial distribution of mean $v_d$.

The minimal impact that results from using LAI that matches the time of observation is not unexpected, since the
meteorological and land cover information from a 2°×2.5° grid cell may not be representative of the typical footprint of a site
measurement (on the order of $10^{-3}$ to $10^1$ km$^2$, e.g. Chen et al., 2009, 2012). This problem has also been highlighted in
previous evaluation efforts in global-scale CTMs (Hardacre et al., 2015; Silva and Heald, 2018). Furthermore, the sample
sizes for all land types are small (N ≤ 16) and the evaluation may be further compromised by inherent sampling biases.

In addition to the evaluation against field observation, we find good correlation (R$^2$ = 0.94) between the annual mean $v_d$ from
GEOS-Chem at 2013 and the 30-year mean $v_d$ of W98 run with static LAI. Overall, our evaluation shows that the quality of
our offline simulation of dry deposition across the four parameterizations in this work is largely consistent with previous
global modelling evaluation efforts.
**4. Impact of Dry Deposition Parameterization Choice on Long-Term Averages**
Here we summarize the impact that the different dry deposition parameterizations may have on simulations of the spatial
distribution of $v_d$ and on the inferred surface O$_3$ concentrations. We begin by comparing the simulated long-term mean $v_d$
across parameterizations, then use a chemical transport model sensitivity experiment to estimate the O$_3$ impacts.





Figure 2 shows the 30-year July daytime average $v_d$ simulated by W98 over vegetated surfaces (defined as the grid cells with
>50% plant cover), and Figure 3 shows the difference between the W98 and the W98_BB, Z03, Z03_BB predictions
respectively. We first focus on results from July because of the coincidence of high surface $O_3$ level, biospheric activity and
$v_d$ in the Northern Hemisphere (NH), and will subsequently discuss the result for December, when such condition holds for
the Southern Hemisphere (SH). W89 simulates the highest July mean daytime $v_d$ in Amazonia (1.2 to 1.4 cm s$^{-1}$), followed by
other major tropical rainforests, and temperate forests in northeastern US. July mean daytime $v_d$ in other temperate regions in
North America and Eurasia typically range from 0.5 to 0.8 cm s$^{-1}$, while in South American and African savannah, and most
parts of China, daytime $v_d$ is around 0.4 to 0.6 cm s$^{-1}$. In India, Australia, western US, and polar tundra Mediterranean region,
July mean daytime $v_d$ is low (0.2-0.5 cm s$^{-1}$) which could be due to either the high temperature or the sparsity of vegetation (or
a combination of both).

The other three parameterizations (W98_BB, Z03, Z03_BB) simulate substantially different spatial distributions of daytime
$v_d$. In North America, we find W98_BB, Z03 and Z03_BB produce lower $v_d$ (by -0.1 to -0.4 cm s$^{-1}$) compared to W98 in
deciduous forest-dominated northeastern US and slightly higher $v_d$ in boreal forest-dominated regions of Canada. Z03 and
Z03_BB produce noticeably lower $v_d$ (by up to -0.2 cm s$^{-1}$) in arctic tundra and grasslands in western US. In southeastern US,
W98_BB and Z03_BB simulate a slightly higher $v_d$ (by up to +0.1 cm s$^{-1}$), while Z03 suggests a slightly lower $v_d$ (by up to -
0.1 cm s$^{-1}$). W98_BB simulates a lower (-0.1 to -0.4 cm s$^{-1}$) $v_d$ in tropical rainforests, with larger reductions concentrated in
southern Amazonia, where July is within the dry season, while the northern Amazonia is not (Malhi et al., 2008). Z03 and
Z03_BB simulate much smaller (-0.4 to -0.6 cm s$^{-1}$) $v_d$ in all tropical rainforests.

Over the midlatitudes in Eurasia, Australia and South America except Amazonia, W98_BB, Z03 and Z03_BB generally
simulate a lower daytime $v_d$ by up to 0.25 cm s$^{-1}$, possibly due to the dominance of grasslands and deciduous forests, where
W98 tends to be more high-biased than other parameterizations when compared to the observations of $v_d$. In southern African
savannah, W98_BB and Z03_BB suggest a much lower daytime $v_d$ (by -0.1 to -0.4 cm s$^{-1}$) because of explicit consideration of
soil moisture limitation to $A_n$ and $g_s$. Z03_BB simulates a particularly high daytime $v_d$ over the high-latitude coniferous forests
(+0.1 to +0.3 cm s$^{-1}$). W98_BB and Z03_BB produce higher daytime daytime $v_d$ (up to +0.15 cm s$^{-1}$) in India and South China
due to temperature acclimation (Kattge and Knorr, 2007), which allows more stomatal opening under the high temperature
that would largely shut down the stomatal deposition in W98 and Z03, as long as the soil does not desiccate. This is guaranteed
by the rainfall from summer monsoon in both regions. Low $v_d$ is simulated by Z03 and Z03_BB in the grasslands near Tibetan
plateau because the grasslands are mainly mapped to tundra land type, which typically has low $v_d$ as discussed in section 3.

Our results suggest that the global distribution of simulated mean $v_d$ depends substantially on the choice of dry deposition
parameterization, driven primarily by the response to hydroclimate and land type-specific parameters, which could impact the





spatial distribution of surface ozone predicted by chemical transport models. To estimate the impact on surface ozone of an
individual parameterization "*i*" compared to the W98 predictions (which we use as a baseline), we apply the following
equation:

$$\Delta O_{3,i} \approx \beta \frac{\overline{v_{d,i}}}{\overline{v_{d_{W98}}}} \quad (3)$$

where $\Delta O_{3,i}$ is the estimated impact on simulated $O_3$ concentrations in a grid box, $\Delta \overline{v_{d,i}}$ is the difference between
parameterization $i$ and W98 simulated mean daytime $v_d$ in that grid box, $\overline{v_{d_{W98}}}$ is W98 output mean daytime $v_d$ for that grid
box, and $\beta$ is the sensitivity of surface ozone to $v_d$ calculated by the method outlined in Section 2.3

Figure 4 shows the resulting estimates of $\Delta O_3$ globally. We find $\Delta O_3$ is the largest in tropical rainforests for all the
parameterizations (up to 5 to 8 ppbv), which agrees with the result from Centoni (2017). Other hotspots of substantial
differences are boreal coniferous forests, eastern US, continental Europe, Eurasian steppe and the grassland in southwestern
China, where $\Delta O_3$ is either relatively large or the signs disagree among parameterizations. In India, Indochina and South China,
$\Delta O_3$ is relatively small but still reaches up to up to -2 ppbv. We find that $\Delta O_3$ is not negligible (1-4 ppbv) in many regions with
relatively high population density, which suggests that the choice of dry deposition parameterization can be relevant to the
uncertainty in the study of air quality and its implication on public health. We note that we have not estimated $\Delta O_3$ for some
regions with low GEOS-Chem-predicted $v_d$ (< 0.25 cm s$^{-1}$, as described in section 2.3), but where the disagreement in $v_d$
between parameterizations can be large (e.g., southern African savannah, see Figure 3). Given this limitation, the impacts on
$O_3$ we have summarized may therefore be spatially conservative.

To explore the importance of seasonality in predictions of $v_d$ and their subsequent impact, we repeat the above analyses for
December. Figure 5 shows the 1982-2011 mean December daytime $v_d$ predicted by W98, while Figure 6 shows the difference
between W98 and the Z03, W98_BB, Z03_BB respectively. High latitudes in the NH are excluded due to the small number of
daytime hours. Z03 and Z03_BB simulate substantially lower in daytime $v_d$ at NH midlatitudes because Z03 and Z03_BB
allow partial snow cover but W98 and W98_BB only allow total or no snow cover. At midlatitudes, the snow cover is not high
enough to trigger the threshold of converting vegetated to snow covered ground in W98 and W98_BB, resulting in lower
surface resistance, and hence higher daytime $v_d$ comparing to Z03 and Z03_BB. In Amazonia, the hotspot of difference in
daytime $v_d$ shifts from the south to the north, which is in the dry season (Malhi et al., 2008). These results for December,
together with our findings from July, suggest that the discrepancy in simulated daytime $v_d$ between W98 and other
parameterizations is due to the explicit response to hydroclimate in the former compared to the latter. Given that field
observations indicate a large reduction of $v_d$ in dry season in Amazonia (Rummel et al., 2007), the lack of dependence of
hydroclimate can be a drawback of W98 in simulating $v_d$ in Amazonia.





Figure 7 shows the resulting estimates of $\Delta O_3$ globally for December using Equation 3.  In all major rainforests, $\Delta O_3$ is smaller
in December due to generally lower sensitivity compared to July. A surprising hotspot of both daytime $\Delta v_d$ and $\Delta O_3$ is the
rainforest/tropical deciduous forest in Myanmar and its eastern bordering region, which also has distinct wet and dry season.
The proximity of December to the dry season, which starts at January (e.g. Matsuda et al., 2005), indicates that the consistent
$\Delta v_d$ between W98 and other parameterizations is driven by hydroclimate as in Amazonia. Comparison with field measurements
(Matsuda et al., 2005) suggests that the W98_BB and Z03_BB capture daytime $v_d$ better than W98, while Z03 may
overemphasize the effect of such dryness. The above reasoning also explains some of the $\Delta v_d$ in India and south China across
the three parameterizations. These findings identify hydroclimate as a key driver of process uncertainty of $v_d$, and therefore its
impact on the spatial distribution of surface ozone concentrations, independent of land type-based biases.

Overall, these results demonstrate that the discrepancy in the spatial distribution of simulated mean daytime $v_d$ resulting from
choice of dry deposition parameterization can have an important impact on the global distribution of surface $O_3$ predicted by
chemical transport models. We find that the response to hydroclimate by individual parametrization not only affects the mean,
but also the seasonality, of predicted surface $O_3$, which is complementary to the findings of Kavassalis and Murphy (2017)
that mainly focus on how shorter-term hydrometeorological variability may modulate surface $O_3$ through dry deposition.

**5. Impact of Dry Deposition Parameterization Choice on Trends and Interannual Variability**
Here we explore the impact that different dry deposition parameterizations may have on predictions of IAV and trends in $v_d$
and on the inferred surface $O_3$ concentrations. We use Theil-Sen method (Sen, 1968) to estimate trends in July daytime $v_d$ (and
any underlying meteorological variables), and use p-value < 0.05 to estimate significance.

Figure 8 shows the trend in July mean daytime $v_d$ from 1982-2011 predicted by each of the parameterizations and scenarios
([Clim], [Clim + LAI], and [Clim + LAI + $CO_2$]). Figure 9 shows the potential impact of these trends in $v_d$ on July daytime
surface ozone, which we estimate to a first order using the following equation:
$$\Delta O_{3_{30y,i}} \approx \beta \times (\text{Annual \% change in } v_{d,i}) \times 30 \; years \; (4)$$
where $\Delta O_{3 \, 30y,i}$ is the absolute change in ozone inferred to a first order as a result of the trend $v_d$ for parameterization $i$ over the
30-years (1982-2011).

In [Clim] simulations (where LAI is held constant), the trend of July daytime $v_d$ is either small or non-significant over the vast
majority of the NH. An exception is observed in the region of Mongolia, where significant increasing trend in $T$ (warming)
and decreasing trend in $RH$ (drying) detected in the MERRA-2 surface meteorological field in July daytime results in
significant decreasing trends using the Z03, W98_BB and Z03_BB parameterizations. This trend is not present in the W98





parameterization as this formulation does not respond to the long-term drying. We find some decreasing trends in $v_d$ across
parts of central Europe and the Mediterranean to varying degrees across the parameterizations. In the SH, we find consistent
decreasing trends across all four parameterizations in southern Amazonia and southern African savannah due to warming and
drying, which we estimate could produce a concomitant increase in July mean surface ozone of between 1 to 3 ppbv (Figure

427    9).


In [Clim+LAI] scenario, all four parameterizations simulate a significant increasing trend of $v_d$ over high latitudes, which is
consistent with the observed greening trend over the region (Zhu et al., 2016). We estimate this could produce a concomitant
increase in July mean surface ozone of between 1 to 3 ppbv. The parameterizations generally agree in terms of the spatial
distribution of these trends in $O_3$. Exceptions include a steeper decreasing trend in most of Siberia predicted by W98, while
the trend is more confined in the eastern and western Siberia in the other three parameterizations. Including the effect of $CO_2$-
induced stomatal closure ([Clim+LAI+CO$_2$] runs) partially offset the increase of $v_d$ in high latitudes, but does not lead to large
changes in both the magnitudes and spatial patterns of $v_d$ trend. We find negligible trends in daytime $v_d$ for December in all
cases. These results show that across all dry deposition model parameterizations, LAI and climate, more than increasing $CO_2$,
can potentially drive significant long-term changes in $v_d$ and should not be neglected when analyzing the long-term change in
air quality over 1982-2011. We note that the importance of the $CO_2$ effect could grow in the coming decades, since the
sensitivity of stomatal conductance to atmospheric $CO_2$ may increase (Franks et al., 2013).

We go on to explore the impact of parameterization choice in calculations of IAV in $v_d$. Figure 10 shows the coefficient of
variation of linearly detrended July daytime $v_d$ ($CV_{vd}$). Figure 11 shows the potential impact this has on IAV in surface ozone,
which we estimate to a first order by the following equation:
$$\sigma_{O_{3,i}} \approx \beta \times CV_{v_{d,i}} \quad (5)$$
where $\sigma_{O3,i}$ is the estimated interannual standard deviation in surface ozone resulting from IAV in $v_d$ given predicted by dry
deposition parameterization $i$. In both cases, we show only the [Clim] and [Clim+LAI] runs, since IAV in $CO_2$ has negligible
impact on interannual variability in $v_d$.

Using the W98 parameterization, IAV in predicted $v_d$ and $O_3$ is considerably smaller in the [Clim] run than that for the [Clim
+ LAI] run, since both the stomatal and non-stomatal conductance in W98 are strong functions of LAI rather than
meteorological conditions. This implies that long-term simulations with W98 and constant LAI can potentially underestimate
the IAV of $v_d$ and surface ozone. In contrast, IAV in $v_d$ calculated by the Z03 parameterization is nearly the same for the [Clim]
and [Clim+LAI] runs. In Z03, $g_s$ is also directly influenced by $VPD$ in addition to temperature and radiation, and non-stomatal
conductance in Z03 is much more dependent on meteorology than W98, leading to high sensitivity to climate. Though the
Ball-Berry model also responds to meteorological conditions, it considers relatively complicated $A_n$-$g_s$ regulation and includes
temperature acclimation, which could dampen its sensitivity to meteorological variability compared to the direct functional



dependence on meteorology in the Z03 multiplicative algorithm. Thus, the climate sensitivity of W98_BB and Z03_BB is in
between Z03 and W98, as is indicated by more moderate difference between $\sigma_{O3,i}$ from [Clim] and [Clim+LAI] runs in Figure

459 11.


For regional patterns of $CV_{vd}$ and $\sigma_{O3}$, we focus on the [Clim+LAI] runs (Fig. 10e to 10h and Fig. 11e to 11h) as it allows for
a comparison of all 4 parameterizations and contain all the important factors of controlling $v_d$. In North America, we estimate
modest IAV in $v_d$ across all 4 parameterizations ($CV_{vd} < 15\%$) in most places. We find this results in relatively low $\sigma_{O3}$ in
northeastern US, and larger $\sigma_{O3}$ in central and southeast US (in the range of 0.3 to 2 ppbv). These results are of a similar
magnitude to the standard deviation of summer mean background ozone suggested by Fiore et al. (2014) over similar time
period, confirming that IAV of dry deposition can be a potentially important component of the natural IAV of surface ozone
in summer over North America.

All parameterizations produce larger $CV_{vd}$ (and therefore larger $\sigma_{O3}$) in southern Amazonia compared to northern and central
Amazonia, but we find substantial discrepancies across parameterizations. The estimated impact on IAV in $O_3$ ($\sigma_{O3}$) in southern
Amazonia ranges from less than 1 ppbv predicted by the W98 and W98_BB parameterizations, to exceeding 1.5 - 2.5 ppbv
predicted by the Z03 parameterization. IAV is also relatively large in central Africa. We find that the parameterizations which
include a Ball-Berry formulation (W98_BB and Z03_BB) estimate higher IAV in this region (with $\sigma_{O3}$ varying between 1 to
4 ppbv), compared to the W98 and Z03 parameterizations ($\sigma_{O3}$ up to 2ppbv). We also note that the Ball-Berry formulations
show more spatial discontinuities compared to W98 and Z03. In our implementation of the Ball-Berry model, impact of soil
moisture on $g_s$ is parameterized as a function of root-zone soil matric potential, which makes $g_s$ very sensitive to variation in
soil wetness when the its climatology is near the point that triggers limitation on $A_n$ and $g_s$. Given the large uncertainty in soil
data (Folberth et al., 2016), such sensitivity could be potentially artificial, which should be taken into consideration when
implementing Ball-Berry parameterizations in large-scale models despite their relatively good performance in site-level
evaluation.

Across Europe, the magnitude of IAV predicted by all four parameterizations show relatively good spatial consistency.
Simulated $CV_{vd}$ is relatively low in western and northern Europe (<10%), which we estimate translates to less than 1 ppbv of
$\sigma_{O3}$. We find larger $CV_{vd}$ (and therefore large $\sigma_{O3}$) over parts of southern Russia and Siberia ($\sigma_{O3}$ up to 2.5 ppbv) from all
parameterizations except W98. The local geographic distribution of $CV_{vd}$ and $\sigma_{O3}$ also significantly differs among the
parameterizations. Z03 and Z03_BB simulate larger $CV_{vd}$ in eastern Siberia than W98_BB, while W98 BB and Z03_BB predict
larger $CV_{vd}$ over the southern Russian steppe then Z03. Finally, all four parameterizations estimate relatively low $CV_{vd}$ and $\sigma_{O3}$
in India, China and Southeast Asia.





We compare the simulated IAV of $v_d$ from all four deposition parameterizations with those recorded by publicly available
long-term observations. The IAV predicted by all four parameterizations at Harvard Forest is between 3% to 7.9%, which is 2
to 6 times lower than that presented in the observations (19%) by Clifton et al. (2017). We find similar underestimates by all
four parameterizations compared to the long-term observation from Hyytiala (Junninen et al., 2009; Keronen et al., 2003;
https://avaa.tdata.fi/web/smart/smear/download), where observed $CV_{vd}$ (11%) is significantly higher than that predicted by the
deposition parameterizations (3.5% - 7.1%). In Blodgett Forest, where $O_3$ uptake is more controlled by gas-phase reactions
(Fares et al., 2010; Wolfe et al., 2011), we find that the models underestimate the observed annual $CV_{vd}$ more seriously (~1%
– 3% compared to 12% in the observations). This suggest that the IAV of $v_d$ may be underestimated across all deposition
parameterizations we investigated (and routinely used in simulations of chemical transport). Clifton et al. (2017) attribute this
to the IAV in non-stomatal deposition, while acknowledging the obscurity of the mechanisms driving such variability, implying
the difficulty in reproducing the observed IAV by existing parameterizations. The scarcity of long-term ozone flux
measurements (Fares et al., 2010, 2017; Munger et al., 1996; Rannik et al., 2012) limits our ability to benchmark the IAV in
our model simulations with observational datasets.

In summary, when both the variability in LAI and climate are considered, the IAV in simulated $v_d$ translates to IAV in surface
$O_3$ of 0.5 – 2ppbv in July for most region. Such variability is predicted to be particularly strong in southern Amazonian and
central African rainforest, where the predicted IAV in July surface $O_3$ due to dry deposition can be as high as 4 ppbv. This
suggests that IAV of $v_d$ can be an important part of the natural variability of surface $O_3$. The estimated magnitude of IAV is
also dependent of the choice of $v_d$ parameterization, which highlights the importance of $v_d$ parameterization choice on
modelling IAV of surface $O_3$.
**6 Discussion and Conclusion**
We present the results of multidecadal global modelling of ozone dry deposition using four different ozone deposition
parameterizations that are representative of the major types of approaches of gaseous dry deposition modelling used in global
chemical transport models. The parameterizations are driven by the same assimilated meteorology and satellite-derived LAI,
which minimizes the uncertainty of model input across parameterization and simplifies interpretation of inter-model
differences. The output is evaluated against field observations and shows satisfactory performance. One of our main goals was
to investigate the impact of dry deposition parameterization choice on long-term averages, trends, and IAV in $v_d$ over a
multidecadal timescale, and estimate the potential concomitant impact on surface ozone concentrations to a first order using a
sensitivity simulation approach driven by the GEOS-Chem chemical transport model.

We find that the performance of the four dry deposition parameterizations against field observations varies considerably over
land types, and these results are consistent with other evaluations, reflecting the potential issue that dry deposition





parameterizations can often be overfit to a particular set of available observations, requiring caution in their application at
global scales. We also find that using more ecophysiologically realistic output $g_s$ predicted by the Ball-Berry model can
generally improve model performance, but at the cost of high sensitivity to relatively unreliable soil data. However, the number
of available datasets of ozone dry deposition observation are still small and concentrated in North America and Europe. We
know of only one multi-season direct observational record in Asia and none in Africa, where air quality can be an important
issue. To better constraint regional dry deposition, effort must be made in making new observations of gaseous dry deposition
(Fares et al., 2017) especially in the under-sampled regions. We also find that many existing ozone flux measurements are not
usable for our evaluation purposes, since only $F_{O3}$ is reported in detail instead of $v_d$. Evaluation and development of ozone dry
deposition parameterizations would be greatly benefited if result of ozone flux measurements is reported in both $F_{O3}$ and $v_d$,
or even have publically available ozone flux and other related micrometeorological variables, which allows both direct
evaluation of $v_d$ and solves the mismatch between coarse model grids and the site (e.g. Wu et al., 2011, 2018).

We find substantial disagreement in the spatial distribution between the mean daytime $v_d$ predicted by the different
parameterizations we tested. We find that these discrepancies are in general a function of both location and season. In NH
summer, $v_d$ simulated by the 4 parameterizations are considerably different in many vegetation-dominated regions over the
world. We estimate that this could lead to around 2 to 5 ppbv in uncertainty of surface ozone concentration simulations over
a vast majority land in the NH. In tropical rainforests, where leaf wetness is prevalent and the dry-wet season dynamics can
have large impact on $v_d$ (Rummel et al., 2007),  we estimate the uncertainty due to dry deposition model choice could even
lead to an uncertainty in surface ozone of up to 8 ppbv. We also find noticeable impacts in parameterization choice during
SH summer, but we note that due to the unreliability of $\beta$ at low $v_d$, we have not assessed its impact on surface ozone in
many high-latitude regions of the NH. In general, we find hydroclimate to be an important driver of the uncertainty. This
demonstrates the potential impact of parameterization choice (or, process uncertainty) of $v_d$ is neither spatiotemporally
uniform nor negligible in most vegetated regions over the world. More multi-seasonal observations are especially needed
over seasonally dry ecosystems where the role of hydroclimate in deposition parameterizations need to be evaluated.
Recently, standard micrometeorological measurements have been used to derive $g_s$ and stomatal deposition of $O_3$ over North
America and Europe (Ducker et al., 2018), highlighting the potential of using global networks of micrometeorological
observation (e.g. FLUXNET (Baldocchi et al., 2001)) to benchmark and calibrate $g_s$ of dry deposition parameterizations,
which could at least increase the spatiotemporal representativeness, if not the absolute accuracy, of dry deposition
parameterizations.

Over the majority of vegetated regions in the NH, we estimate the IAV of mean daytime $v_d$ is generally on the order of 5 to
15% and may contribute between 0.5 to 2 ppbv of IAV in July surface $O_3$ over the thirty-year period considered here, with
each parameterization simulating different geographic distribution of where IAV is highest. The predicted IAV from all four
models is smaller than what long-term observations suggest, but is still comparable to the long-term variability of background




ozone over similar timescales in U.S. summer (Brown-Steiner et al., 2018; Fiore et al., 2014). This would seem to confirm that
$v_d$ may be a substantial contributor to natural IAV of $O_3$ in summer, at least in U.S. In the southern Hemisphere, the IAV
mainly concentrates in drier part of tropical rainforests. The Ball-Berry parameterizations simulate large and spatially
discontinuous $CV_{vd}$ and $\sigma_{O3}$ due to their sensitivity to soil wetness. Globally, we find that IAV of $v_d$ in W98 is mostly driven
by LAI, while in other parameterizations climate generally plays a more important role. We therefore emphasize that temporal
matching of LAI is important for consistency when W98 is used in long-term simulations. The scarcity of long-term ozone
deposition measurement poses significant difficulty in evaluating the model predictions over interannual (and in particular
multidecadal) timescales. This information is helpful in designing and identifying sources of error in model experiments that
involve variability of $v_d$.

We are also able to detect statistically significant trends in July daytime $v_d$ over several regions. The magnitudes of trend are
up to 1% per year and both climate and LAI contribute to the trend. All four deposition parameterizations identify three main
hotspots of decreasing July daytime $v_d$ (southern Amazonia, southern African savannah, Mongolia), which we link mainly to
increasing surface air temperature and decreasing relative humidity. Meanwhile, extensive areas at high latitudes experience
LAI-driven increasing July daytime $v_d$, consistent with the greening trend in the region (Zhu et al., 2016). We don't find a
strong influence of $CO_2$-induced stomatal closure in the trend over this time period. Over the 30-years we estimate the trend
in July daytime $v_d$ could translate approximately to 1 to 3 ppbv of ozone changes in the areas of impact, indicating the potential
effect of long-term changes in $v_d$ on surface ozone. This estimate should be considered conservative, since we are unable to
reliably test the sensitivity of ozone to regions with low $v_d$ with our approach.

While the approach we have presented here allows us to explore the role of dry deposition parameterization choice on
simulations of long-term means, trends, and IAV in ozone dry deposition velocity, there remain some limitations and
opportunities for development. First, we only used one LAI and assimilated meteorological product. The geographic
distribution of trend and IAV of $v_d$ may vary considerably as the LAI and meteorological products used due to their inherent
uncertainty (e.g. Jiang et al., 2017). While we expect the qualitative conclusions about how LAI and climate controls the
modelled trend and IAV of $v_d$ to be robust to the choice of data set, the magnitude and spatial variability could be affected.
Second, the estimated effects on surface $O_3$ are a first-order inference based on a linear approximation of the impact that $v_d$
has directly on $O_3$. We have not applied our analysis to regions with low baseline GEOS-Chem $v_d$, where other components of
parameterization (e.g. definition and treatment of snow cover, difference in ground resistance) may have major impact on $v_d$
prediction (Silva and Heald, 2018), nor accounted for the role that $v_d$ variability can have on other chemical species which
would have feedbacks on $O_3$. Moreover, the sensitivity of surface ozone to dry deposition velocity may be dependent on the
choice of chemical transport model (here, the GEOS-Chem model has been used), and possibly the simulation year. Finally,
we have neglected the effect of land use and land cover change on global PFT composition at this stage, which can be another
source of variability for $v_d$. Nevertheless, the relatively high *NMAEF* of simulated $v_d$ and the inherent uncertainty in input data





(land cover, soil property, assimilated meteorology and LAI) are considered as the major source of uncertainty in our
predictions of $v_d$.

The impact of dry deposition parameterization choice may be generalizable to other trace gases with deposition velocity
controlled by surface resistance, and for which stomatal resistance is an important control of surface resistance (e.g. $NO_2$). As
$v_d$ has already been recognized as a major source of uncertainty in deriving global dry deposition flux of $NO_2$ and $SO_2$ (Nowlan
et al., 2014), systematic investigation on the variability and uncertainty of $v_d$ for other relevant chemical species does not only
contribute to understanding the of gaseous dry deposition role on air quality, but also biogeochemical cycle. Particularly,
gaseous dry deposition has been shown to be a major component in nitrogen deposition (Geddes and Martin, 2017; Zhang et
al., 2012), highlighting the potential importance of understanding the role of $v_d$ parameterization in modelling regional and
global nitrogen cycle.

Here we have built on the recent investigations of modelled global mean (Hardacre et al., 2015; Silva and Heald, 2018) and
observed long-term variability (Clifton et al., 2017) of $O_3$ $v_d$. We are able to demonstrate the substantial impact of $v_d$
parameterization on modelling the global mean and IAV of $v_d$, and their non-trivial potential impact on simulated seasonal
mean and IAV of surface ozone. We demonstrate that the parameterizations with explicit dependence on hydroclimatic
variables have higher sensitivity to climate variability than those without. Difficulties in evaluating predictions of $v_d$ for many
regions of the world (e.g. most of Asia and Africa) persist due to the scarcity of measurement. This makes a strong case for
additional measurements (e.g. Kammer et al., 2019; Li et al., 2018; Stella et al., 2011a), empirical studies (e.g. Ducker et al.,
2018) and model-observation integrations (e.g. Silva et al., 2019) of ozone dry deposition at different timescales, which would
be greatly facilitated by an open data sharing infrastructure (e.g. Baldocchi et al., 2001; Junninen et al., 2009).
**Code Availability**
The source code and output of the dry deposition parameterizations can be obtained by contacting the corresponding author
(jgeddes@bu.edu).
**Appendix**

|  | W98 | Z03 | W98_BB | Z03_BB |
|---|---|---|---|---|
| $R_a$ | $R_a = \frac{1}{\kappa u_*}\left[\ln(\frac{z}{z_0}) - \Psi\left(\frac{z}{L}\right) + \Psi(\frac{z_0}{L})\right]$<br><br>When $\varsigma \geq 0, \Psi(\varsigma) = -5\varsigma$<br><br>When $\varsigma < 0, \Psi(\varsigma) = 2\ln(\frac{1+\sqrt{1-16\varsigma}}{2})$ |  |  |  |





| $R_b$ | $R_b = \dfrac{2}{\kappa u_*}\left(\dfrac{Sc}{Pr}\right)^{2/3}$ | | | |
|---|---|---|---|---|
| $R_s$ | $R_s = r_s(PAR,LAI)f_T\dfrac{D_{H_2O}}{D_{O_3}}$ | $R_s = \dfrac{r_s(PAR,LAI)}{(1-w_{st})f_T f_{vpd} f_\psi}\dfrac{D_{H_2O}}{D_{O_3}}$ | $g_s = g_0 + m\dfrac{A_n}{C_s}h_s$  $R_s = \dfrac{1}{g_s}\dfrac{D_{H_2O}}{D_{O_3}}$ | $g_s = g_0 + m\dfrac{A_n}{C_s}h_s$  $R_s = \dfrac{1}{(1-w_{st})g_s}\dfrac{D_{H_2O}}{D_{O_3}}$ |
| Cuticular Resistance ($R_{cut}$) | $R_{cut} = \dfrac{R_{cut_0}}{LAI}$ | For dry surface,  $R_{cut} = \dfrac{R_{cut_{d0}}}{e^{0.03RH}LAI^{0.25}u_*}$  For wet surface,  $R_{cut} = \dfrac{R_{cut_{w0}}}{LAI^{0.5}u_*}$ | Same as W98 | Same as Z03 |
| In-canopy aerodynamic resistance ($R_{ac}$) | Prescribed | $R_{ac} = R_{ac_0}\dfrac{LAI^{0.25}}{u_*}$ | | |
| Ground Resistance ($R_g$) | Prescribed | | | |
| Lower-canopy aerodynamic resistance ($R_{alc}$) | $R_{alc} = 100\left(1+\dfrac{1000}{R+10}\right)$ | - | | |
| Lower-canopy surface resistance ($R_{clc}$) | Prescribed | - | | |

**Table A1:** Brief description of the four dry deposition parameterizations. $\kappa$ = von Karman constant, $u_*$ = friction velocity, $z$ = reference height, $z_0$ = roughness length, $L$ = Obukhov length, $Sc$ = Schmidt's number, $Pr$ = Prandtl number for air, $LAI$ = leaf area index, $PAR$ = photosynthetically active radiation, $D_x$ = Diffusivity of species $x$ in air, $f_T$ = temperature ($T$) stress function, $f_{vpd}$ = vapour pressure deficit ($VPD$) stress function, $f_\psi$ = leaf water potential ($\psi$) stress function, $w_{st}$ = stomatal blocking fraction, $A_n$ = Net photosynthetic rate, $g_0$ = minimum stomatal conductance, $m$ = Ball-Berry slope, $C_s$ = $CO_2$ concentration on leaf surface, $h_s$ = relative humidity on leaf surface, $RH$ = relative humidity, $h$ = canopy height, $R$ = downward shortwave radiation

| CLM PFT | Z03 surface type |
|---|---|
| Needleleaf evergreen tree - temperate | Evergreen needleleaf trees |
| Needleleaf evergreen tree - boreal | |
| Needleleaf deciduous tree - boreal | Deciduous needleleaf trees |





| Broadleaf evergreen tree - tropical | Tropical broadleaf trees |
| Broadleaf deciduous tree - tropical | Deciduous broadleaf trees |
| Broadleaf deciduous tree - temperate | |
| Broadleaf deciduous tree - boreal | |
| Broadleaf evergreen shrub - temperate | Thorn shrubs |
| Broadleaf deciduous shrub - temperate | Deciduous shrubs |
| Broadleaf deciduous shrub - boreal | |
| $C_3$ arctic grass | Tundra |
| $C_3$ grass | Short grass |
| $C_4$ grass | Corn[*] |
| $C_3$ crop | Crops |

**Table A2:** Mapping between CLM PFT and Z03 surface type.
*$C_4$ grasses are mapped to corn due to the similarity in photosynthetic pathway, and hence stomatal control

| Land Type | Longitude | Latitude | Season | Mean daytime $v_d$ (cm s$^{-1}$) | Citation |
|---|---|---|---|---|---|
| Deciduous Forest | -80.9° | 44.3° | Summer | 0.92 | Padro et al., 1991 |
| | | | Winter | 0.28 | |
| | 99.7° | 18.3° | Spring | 0.38 | Matsuda et al., 2005 |
| | | | Summer | 0.65 | |
| | -72.2° | 42.7° | Summer | 0.61 | Munger et al., 1996 |
| | | | Winter | 0.28 | |
| | -78.8° | 41.6° | Summer | 0.83 | Finkelstein et al., 2000 |
| | -75.2° | 43.6° | Summer | 0.82 | |
| Coniferous Forest | -3.4° | 55.3° | Spring | 0.58 | Coe et al., 1995 |
| | -79.1° | 36.0° | Spring | 0.79 | Finkelstein et al., 2000 |
| | -120.6° | 38.9° | Spring | 0.58 | Kurpius et al., 2002 |
| | | | Summer | 0.59 | |
| | | | Autumn | 0.43 | |
| | | | Winter | 0.45 | |
| | -0.7° | 44.2° | Summer | 0.48 | Lamaud et al., 1994 |
| | 105.5° | 40.0° | Summer | 0.39 | Turnipseed et al., 2009 |
| | -66.7° | 54.8° | Summer | 0.26 | Munger et al., 1996 |
| | 11.1° | 60.4° | Spring | 0.31 | Hole et al., 2004 |




| | | | Summer | 0.48 | |
|---|---|---|---|---|---|
| | | | Autumn | 0.20 | |
| | | | Winter | 0.074 | |
| | 8.4° | 56.3° | Spring | 0.68 | Mikkelsen et al., 2004 |
| | | | Summer | 0.80 | |
| | | | Autumn | 0.83 | |
| Tropical Rainforest | 117.9° | 4.9° | Wet | 0.5 | Fowler et al., 2011[#] |
| | | | Wet | 1.0 | |
| | -61.8° | -10.1° | Wet | 1.1 | Rummel et al., 2007 |
| | | | Dry | 0.5 | |
| | -60.0° | 3.0° | Wet | 1.8 | Song-Miao et al., 1990 |
| Grass | -88.2° | 40.0° | Summer | 0.56 | Droppo, 1985 |
| | -3.2° | 57.8° | Spring | 0.59 | Fowler et al., 2001 |
| | | | Summer | 0.56 | |
| | | | Autumn | 0.42 | |
| | -119.8° | 37.0° | Summer | 0.15 | Padro et al., 1994 |
| | -8.6° | 40.7° | Summer | 0.22 | Pio et al., 2000 |
| | | | Winter | 0.38 | |
| | -104.8° | 40.5° | Spring | 0.22 | Stocker et al., 1993 |
| | 10.5° | 52.4° | Spring | 0.44 | Mesźaros et al., 2009 |
| | -96.4° | 39.5° | Summer | 0.62 | Gao and Wesely, 1995 |
| Crops | -2.8° | 55.9° | | 0.69 | Coyle et al., 2009 |
| | -88.4° | 40.1° | | 0.53 | Meyers et al., 1998 |
| | | | | 0.12 | |
| | -87.0° | 36.7° | | 0.85 | |
| | | | | 0.39 | |
| | -86.0° | 34.3° | Not applicable[*] | 0.40 | |
| | -120.7° | 36.8° | | 0.76 | Padro et al., 1994 |
| | 8.0° | 48.7° | | 0.41 | Pilegaard et al., 1998 |
| | 2.0° | 48.9° | | 0.60 | Stella et al., 2011 |
| | 0.6° | 44.4° | | 0.47 | |
| | 1.4° | 43.8° | | 0.37 | |

**Table A3:** Information of all the measurement sites included in model evaluation



*Crops are heavily influenced by management practices rather than natural seasonality. Thus, two data sets in the same location
generally represent before and after certain a crop phenology or human management event.
#The two measurements are taken at a rainforest and an oil palm plantation nearby.

**Author Contributions**

AYHW and JAG developed the ideas behind this study, formulated the methods, and designed the model experiments. AYHW
wrote the dry deposition code and ran the chemical transport model simulations. Data analysis was performed by AYHW, with
input and feedback from JAG. APKT provided the photosynthesis model code, and co-supervised the dry deposition code
development. SJS compiled the dry deposition observations used for evaluation. Manuscript preparation was performed by
AYHW, reviewed by JAG, and commented, edited, and approved by all authors.

**Acknowledgement**

This work was funded by an NSF CAREER grant (ATM-1750328) to project PI J.A. Geddes; and the Vice-Chancellor
Discretionary Fund (Project ID: 4930744) from The Chinese University of Hong Kong (CUHK) given to the Institute of
Environment, Energy and Sustainability. Funding support to SJS was provide by a National Science Foundation grant to C.L.
Heald (ATM-1564495). We also thank the Global Modelling and Assimilation Office (GMAO) at NASA Goddard Flight
Center for providing the MERRA-2 data, Ranga Myneni for GIMMS LAI3g product, Petri Keronen and Ivan Mammarella for
the flux measurements in Hyytiala, Silvano Fares and Allen Goldstein for the flux measurement in Blodgett Forest, and
Leiming Zhang and Zhiyong Wu for the source code of Z03.



















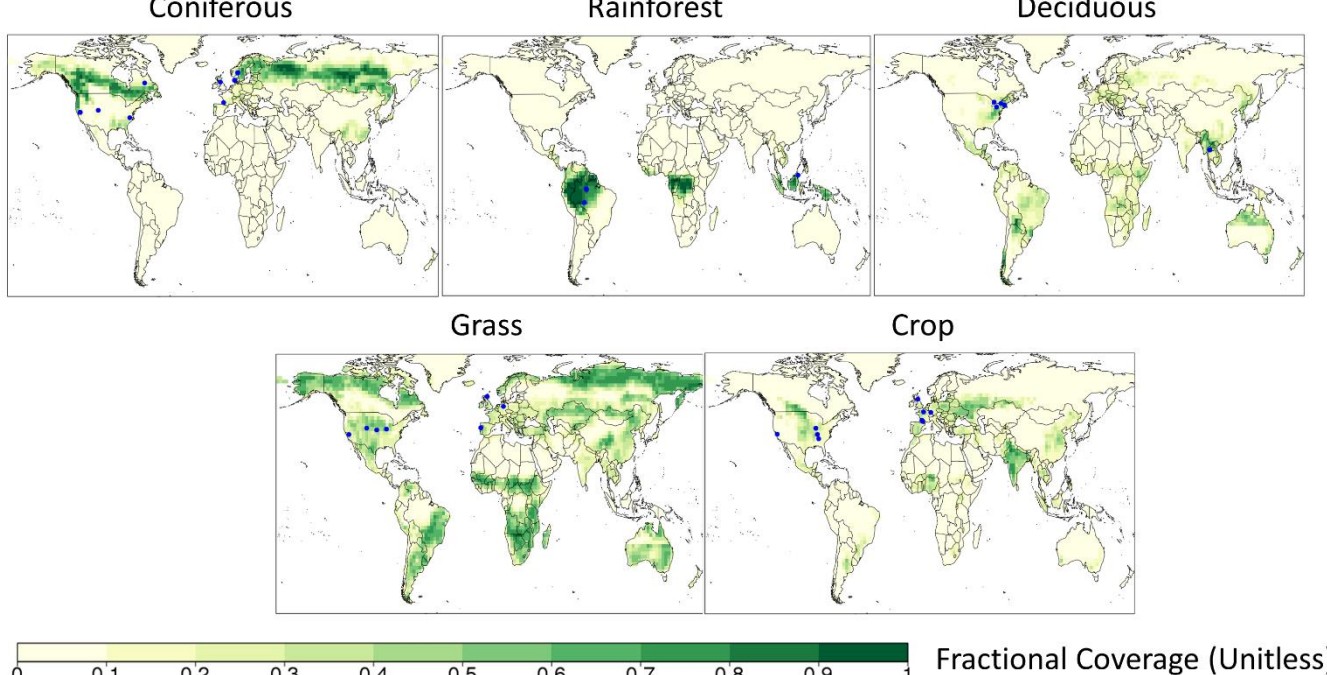


**Figure 1:** Fractional coverage of each major land type at each grid cell. Blue dots indicate the locations of the observational

sites.

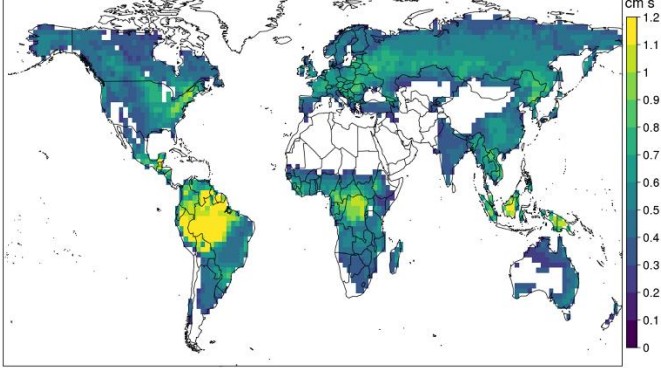


**Figure 2:** 1982-2011 July mean daytime $v_d$ (solar elevation angle > 20°) over vegetated land surface simulated by W98.






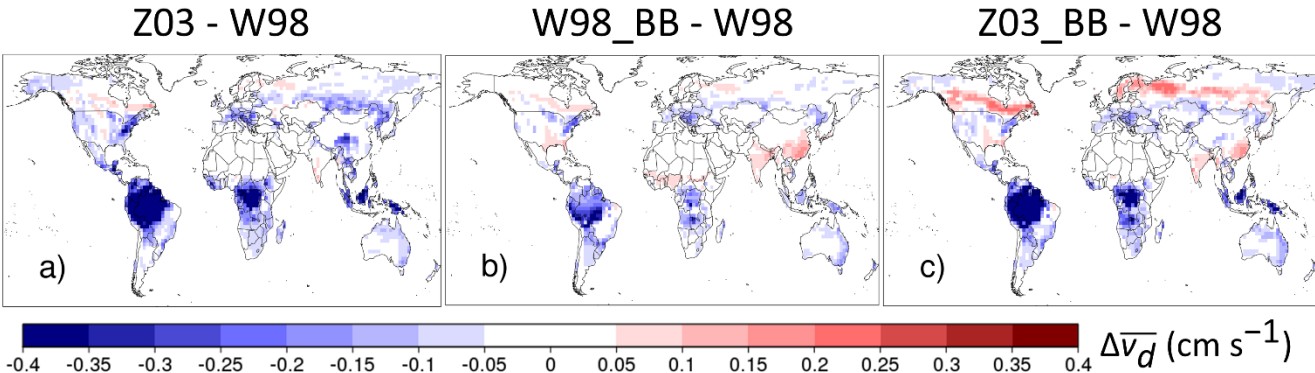


**Figure 3:** Differences of 1982-2011 July mean daytime $v_d$ ($\Delta\overline{v_d}$) between three other parameterizations (Z03, W98_BB and

Z03_BB) and W98 over vegetated land surface.


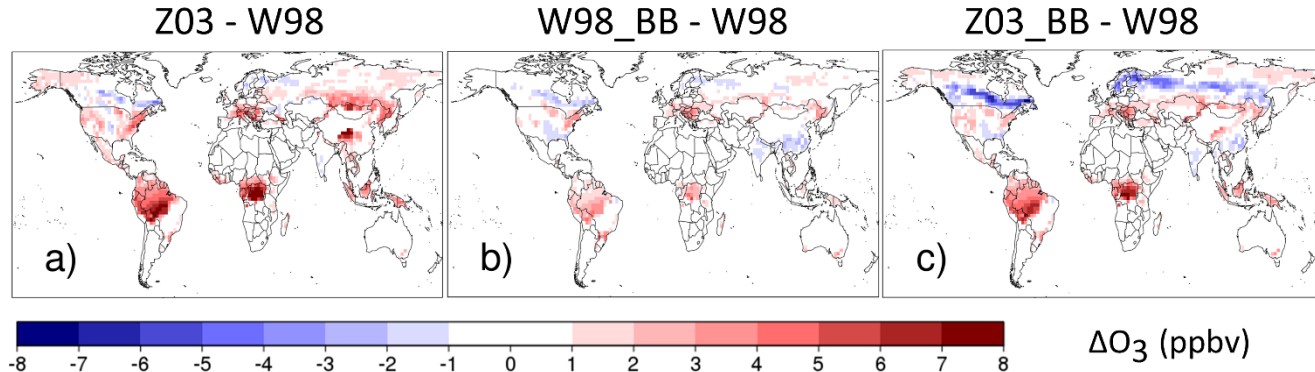


**Figure 4:** Estimated difference in July mean surface ozone ($\Delta O_3$) due to the discrepancy of simulated July mean daytime $v_d$

among the parameterizations.









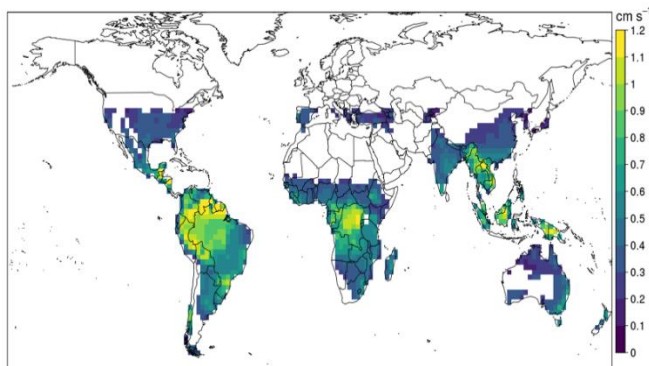


**Figure 5:** 1982-2011 December mean daytime $v_d$ (solar elevation angle > 20°) over vegetated land surface simulated by

W98. The data over high latitudes over Northern Hemisphere is invalid due to insufficient daytime hours over the month (<

100 hours month$^{-1}$)


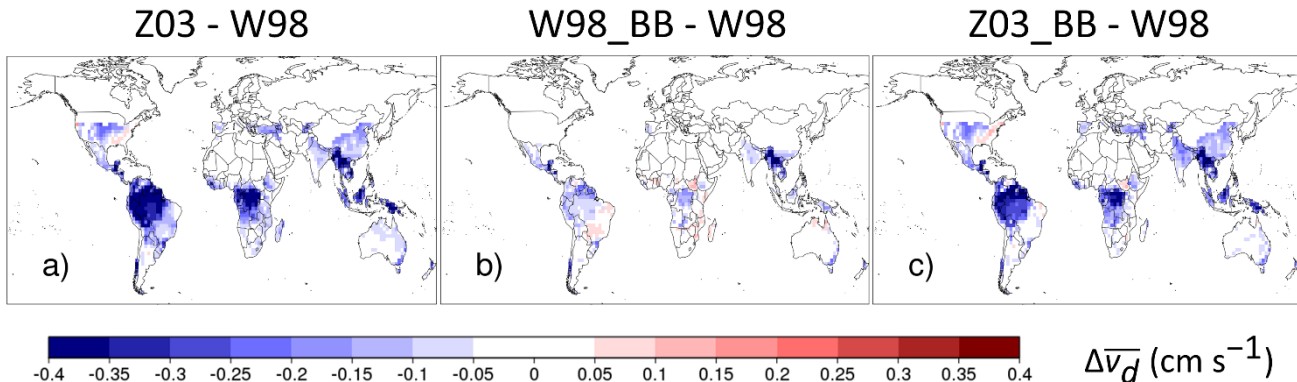


**Figure 6:** Differences of 1982-2011 December mean daytime $v_d$ ($\Delta\overline{v_d}$) between three other parameterizations (Z03, W98_BB

and Z03_BB) and W98 over vegetated land surface.





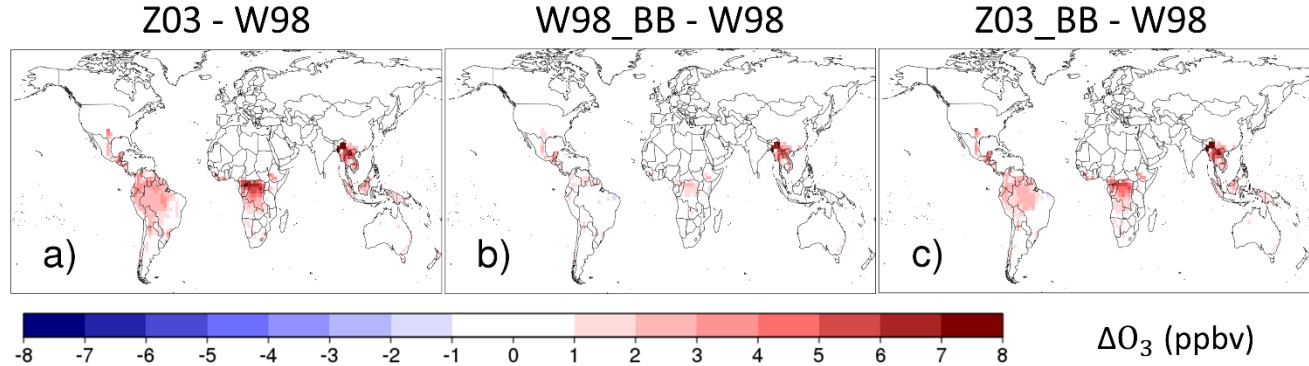

**Figure 7:** Estimated difference in December mean surface ozone ($\Delta O_3$) due to the discrepancy of simulated December mean daytime $v_d$ among the parameterizations.

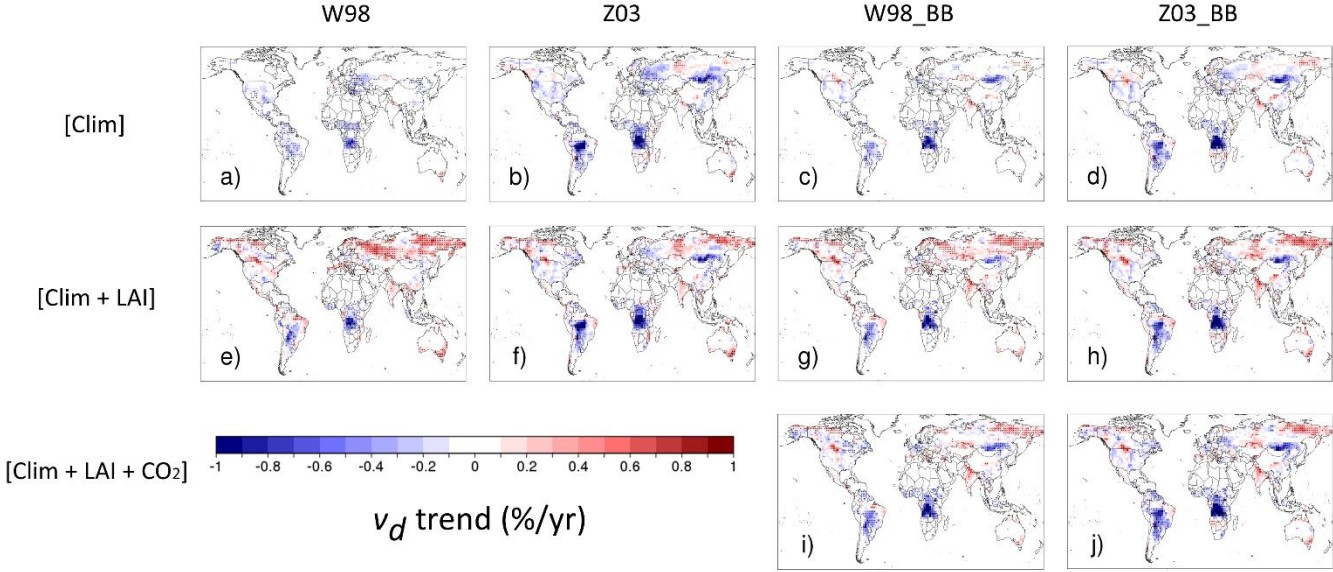

**Figure 8:** Trends of July mean daytime $v_d$ during 1982-2011 over vegetated land surface. Black dots indicate statistically significant trends ($p < 0.05$)

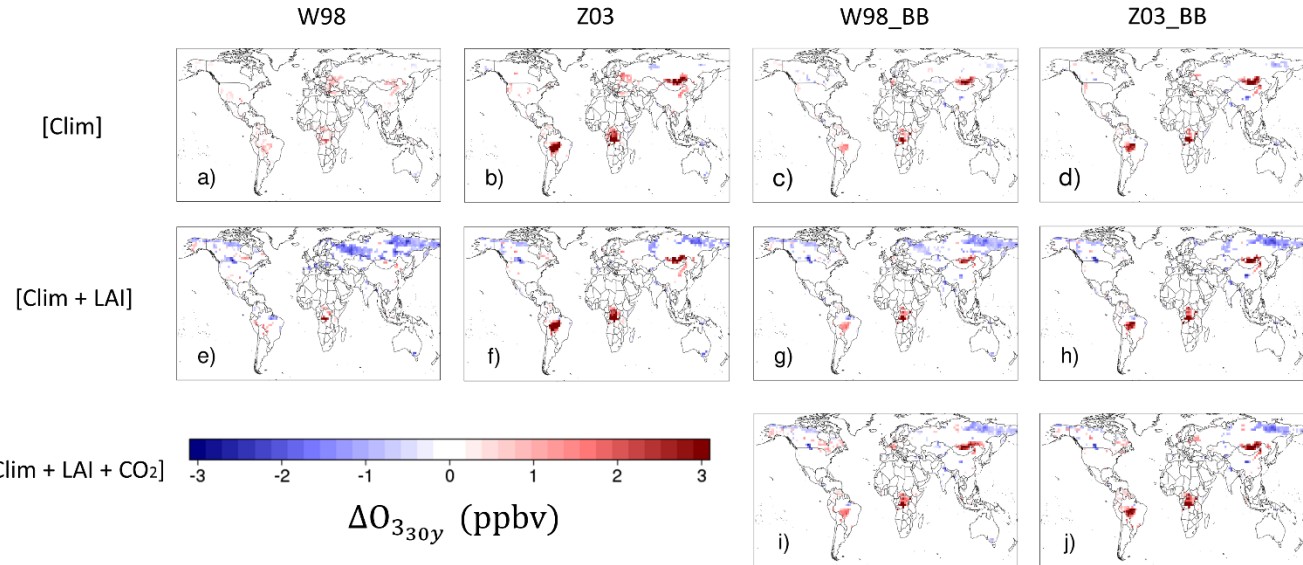

**Figure 9:** Estimated impact of trends of July mean daytime $v_d$ on July mean surface ozone during ($\Delta O_{3\ 30y}$) 1982-2011 over vegetated land surface. Only grid points with statistically significant trends ($p < 0.05$) in July mean daytime $v_d$ are considered.

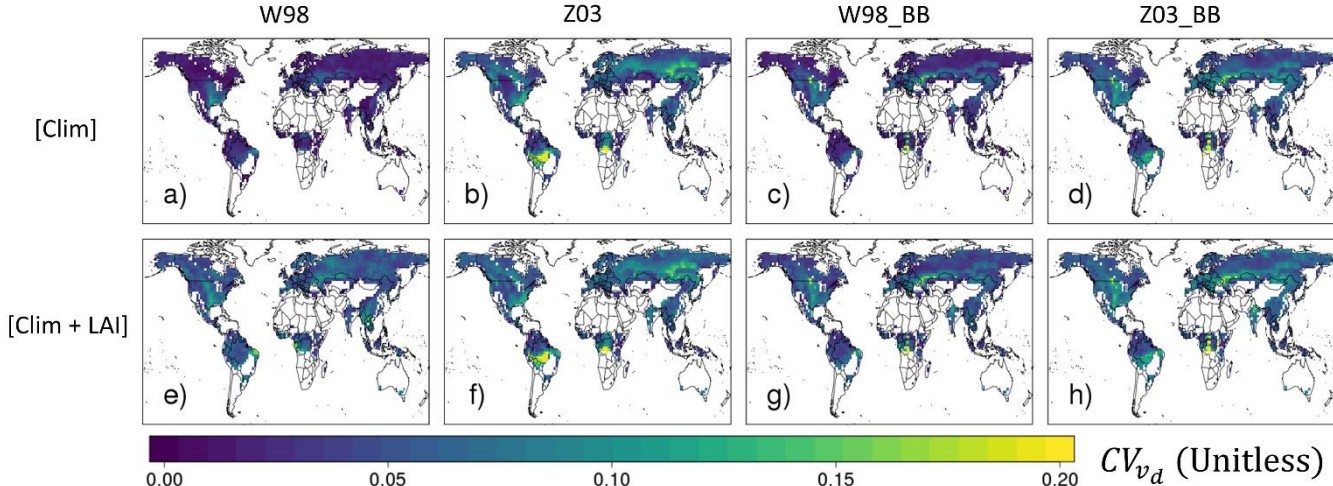

**Figure 10:** Interannual coefficient of variation of linearly detrended July mean daytime $v_d$ ($CV_{vd}$) during 1982-2011 over vegetated land surface.





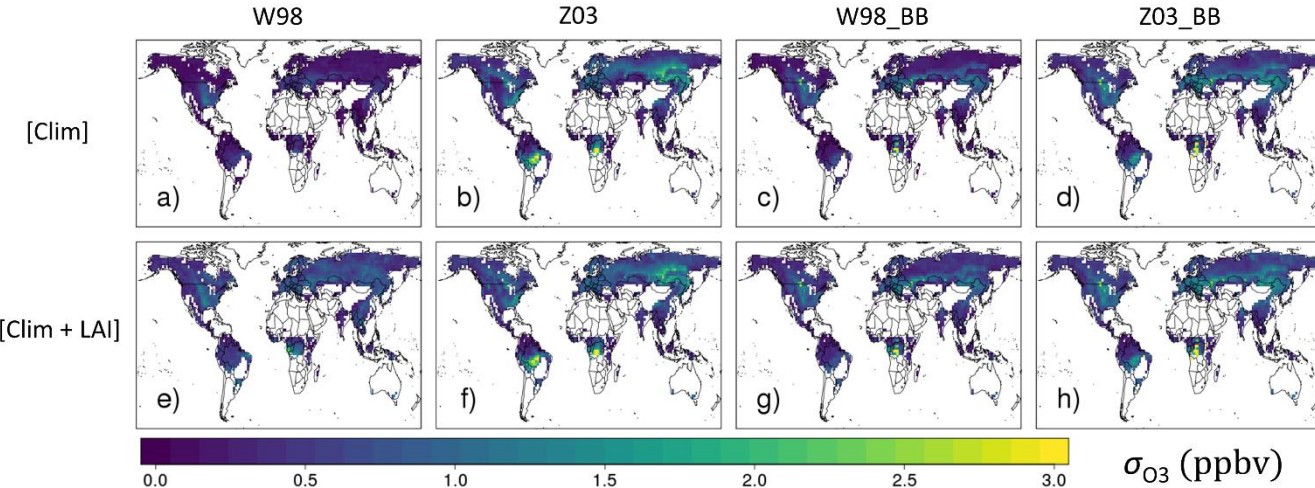

**Figure 11:** Estimated contribution of IAV in July mean daytime $v_d$ to IAV of July mean surface ozone ($\sigma_{O3}$) during 1982-2011 over vegetated land surface.

| $v_d$ simulation | Meteorology | LAI | Atmospheric CO₂ concentration |
|---|---|---|---|
| [Clim] | MERRA-2 meteorology | LAI3g monthly climatology | 390 ppm |
| [Clim+LAI] | | LAI3g monthly time series | |
| [Clim+LAI+CO₂] | | | Manoa Loa time series |

**Table 1:** List of $v_d$ simulations with input data

| Land types | Metrics | Static LAI | | | | Dynamic LAI | | | |
|---|---|---|---|---|---|---|---|---|---|
| | | W98 | Z03 | W89-BB | Z03_BB | W98 | Z03 | W89-BB | Z03_BB |
| Dec | *NMBF* | **0.134** | -0.367 | -0.287 | -0.142 | **0.119** | -0.376 | -0.299 | -0.153 |
| (*N*=8) | *NMAEF* | 0.322 | 0.369 | 0.305 | **0.215** | 0.319 | 0.376 | 0.321 | **0.226** |
| Con | *NMBF* | -0.362 | -0.217 | -0.252 | **-0.025** | -0.355 | -0.209 | -0.248 | **-0.023** |





| | | | | | | | | | |
|---|---|---|---|---|---|---|---|---|---|
| (N=16) | NMAEF | 0.448 | 0.455 | 0.483 | **0.399** | 0.427 | 0.458 | 0.470 | **0.394** |
| Tro | NMBF | **0.080** | -0.808 | -0.086 | -0.438 | **0.075** | -0.813 | -0.090 | -0.441 |
| (N=5) | NMAEF | 0.423 | 0.831 | **0.404** | 0.569 | 0.422 | 0.832 | **0.399** | 0.567 |
| Gra | NMBF | 0.276 | **0.015** | 0.175 | 0.097 | 0.294 | **0.011** | 0.186 | 0.110 |
| (N=10) | NMAEF | 0.392 | 0.479 | **0.307** | 0.318 | 0.396 | 0.467 | **0.302** | 0.311 |
| Cro | NMBF | 0.297 | 0.360 | **0.241** | 0.282 | 0.318 | 0.371 | **0.255** | 0.292 |
| (N=11) | NMAEF | **0.473** | 0.541 | 0.474 | 0.570 | 0.485 | 0.550 | **0.480** | 0.576 |

**Table 2:** Performance metrics (*NMBF* and *NMAEF*) for daytime average $v_d$ simulated by the four dry deposition parameterizations. "Static LAI" is the result from [Clim] run, which uses 1982-2011 AVHRR monthly climatological LAI, while "Dynamic LAI" is the result from [Clim+LAI], which uses 1982-2011 AVHRR LAI time series. Dec = deciduous forest, Con = coniferous forest, Tro = tropical rainforest, Gra = grassland, Cro = cropland. *N* indicates the number of observational datasets involved in that particular land type. The best performing parameterization for each land type has its performance metrics bolded.

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
