# Peer review of "Importance of Dry Deposition Parameterization Choice in Global"

_Atmospheric Chemistry and Physics, 2019_

## Referee Comment (RC1) · Anonymous Referee #1 · 18 Jun 2019

Wong et al. is a well written paper that will be useful for the chemical-transport modeling community as well as the community making ozone flux observations. Wong et al. investigate four ozone dry deposition parameterizations used commonly in global scale chemical transport modeling, including how they compare against observations, how much interannual variability is simulated, and whether there is a trend in ozone deposition velocity over the past 30 years. The authors then examine the impacts of interannual variability and long-term trends in ozone deposition velocity may have influence ozone air quality. This paper is the first to examine such questions at the global scale. One of the paper's strengths is that it suggests that any one of four current parameterizations for ozone dry deposition is not necessarily better than another in terms of capturing observed ozone deposition velocities at several observational sites around the world. Another interesting finding is that even though the interannual variability in simulated ozone deposition velocity is muted compared to three ozone flux datasets with ~10 years of measurements, the simulated interannual variability has implications for simulated surface ozone concentrations.

Major issues
1. the linearity of response of surface ozone concentration to ozone deposition velocity is uncertain, but a major assumption in this study. i'm not convinced that the results from wong et al. 2018 are sufficient to warrant confidence in this assumption. one reason being that they were testing the response to surface ozone to LAI, which involves changes in several processes.
2. the authors' attribution of biases and intermodel differences are entirely speculative. there is no rigorous evaluation of the processes/aspects leading to differences. i tend to not be in favor of such speculation and I think it masks the strength of the model evaluation (that not any one parameterization is best or worst) and model intercomparison.

Minor issues
10: I tend to think the sinks of ozone are chemistry and dry deposition so "second largest sink" doesn't say much to me
15-16: "to drive four ozone dry deposition parameterizations"
62: I wouldn't say Silva & Heald 2018 is a review
66: "account for" is vague; in general this sentence implies canopy column models are better than big-leaf ones, which has yet to be shown in the literature
67: the authors said previously that reaction with BVOCs is a nonstomatal pathway so here saying that it is in addition to surface sinks is a little confusing
67-71: canopy column models still use resistance networks …
77-80: this has yet to be shown… these formulations can be variable across models …
80-88: the connection between these paragraphs (last sentence of previous one and first sentence of next one) could be articulated better
101: Hardacre et al. show factor of 2-3 differences across models – so are all models' seasonal cycles well represented? also I suggest changing "demonstrating" to "suggesting"
125: "unable" seems harsh; it doesn't seem Clifton et al. even tried to do this
128: cut "physics"
145: I find the placement/existence of this sentence strange. the authors don't investigate the same parameterizations that Wu et al. do.
153: refs for strong empirical relationship
162-173: I see that the authors have basically organized their parameterizations according to model (w/ exception of #2)
1) The GEOS Chem parameterization
2) Zhang parameterization
3) The CESM parameterization
4) The UKCA parameterization
I didn't realize this at first and the parameterizations chosen seemed quite strange. I would urge the authors to re-frame their parameterization description (but also noting that their parameterizations are not exact replicates of a given model)
175: It doesn't quite make sense to me that the authors say the Zhang parameterization is "open source" in one sentence and a couple sentences later say that implementing it required personal communication with Zhiyong and Leiming.
180: Given that GEOS Chem doesnt have a land surface model, I think the authors need to clarify how exactly Anet is calculated.
182-183: It's fine not to test Ra and Rb, but i suggest that the authors do not use this qualifier. This isn't well understood (Does Fares et al. even show this?)
188-9: has this model been evaluated? or used previously?
194-5: what are these variables used for?
195: presumably the authors' decisions about land type mapping (& differences for "W89" vs "Z03") impact the authors' results… one implication of this is that the authors' statement in the abstract or introduction that the only thing different across parameterizations is the model structure is not necessarily true
197: i would suggest cutting the "(eg. leaf physiological and soil hydrauilic constants)" – becoming more specific here doesn't help readers when the parameterizations are not laid out and we have no idea what these terms do/stand for
198: what's z0?
198: how is leaf wetness calculated? how is snow calculated?
203: how do the authors scale PFT-specific LAI? is there an established method of doing this? presumably this has implications for the findings
217: i think the authors need to articulate here or in the introduction the various effects that high CO2 may have on ozone dry deposition velocity and the various uncertainties in our understanding of CO2 fertilization (& reference previous work examining this)
229: is the proper/up-to-date way of referencing GEOS-Chem?
237: binned = jargon
243-246: discussing about dry deposition of other species and impacts on ozone requires introducing some concepts (or cutting talking about dry deposition of other species)
249-251: this seems like a strange choice to me. it's not differences in transport per se, it's differences in background ozone caused by changes in ozone dry deposition. why wouldn't the authors want to capture this? because it contributes to nonlinear responses to ozone dry deposition?
249: what is the baseline simulation?
254: Why not CLIM+LAI+CO2 as well?
261-3: How many sites does this cut?
265: Fractional coverage of what? (please spell out in text) Why are these figures shown? they are not very useful for the reader
270-1: Not sure what the point of this sentence is
273: it seems strange to me that the authors would generalize such as bias, given that its unclear if the bias is caused by a particular attribute of an land type or process, and that the land type-specific biases differ across the parameterizations
282: what does N=5 mean? 5 sites? 5 data points?
288: if the authors are implying ambient chemistry is happening then they should just say it
300: meaning that the authors do not leverage it
301-302: I'm not sure that the following lines illustrate this; in other words, i think BB "generally but not universally leads to improvements" is not supported by the actual findings — it seems to be for Z03 — but not for Wesely — which may suggest that we need to be paying attention to nonstomatal deposition estimates too.
313-4: what particular problem has been highlighted?
315: sampling biases meaning that the authors are not evaluating most locations on earth, right? the authors are sampling the time/place of the measurements
317-320: not sure what the point of this paragraph is. what is the hypothesis being investigated?
334-5: recommend that the authors don't speculate here or elsewhere
349-50: on a similar note as the above comment, how do the authors know this?
353: "is not desiccated"?
358: i don't think the authors show this; they just speculate that this is the cause.
368: will the authors more carefully articulate what Centoni finds so that the reader knows how to compare the findings
370: i assume that the authors are identifying the hot spot regions through their large delta O3. related: perhaps the authors are missing a delta on the $v_{d,i}$ in Equation 3.
378: are the authors really "exploring the importance of seasonality in predictions of vd and their subsequent impact" with their current approach? (see comment below for line 404)
382-4: i suggest a semi colon connecting these two sentences
385: "shifts from the south to the north relative to July"
387: i'm not a fan of the authors' use of the term hydroclimate — it's vague — can the authors just say soil moisture or VPD or leaf wetness?
398: the suggestion that "hydroclimate [is] a key driver of process uncertainty" seems limited to the tropics/subtropics. am i correct in this interpretation? if so, this should be emphasized.
404: are the authors actually showing the impacts on seasonality? showing the impact in each season is not the same as showing the impact on seasonality (a couple of easy calculations could help here)
409: briefly describe this method such as the limitations/strengths of it
413: what trends? trends in meteorology, LAI, and/or CO2?
415: how is the annual change in vd estimated? is it using the Theil-Sen method? this part needs better explanation; the reader needs to at least have some concept of what the method used is
423-4: but they are small or nonsignificant per the first line of the paragraph?
439: or it may decrease as plants acclimate or as nutrients become limiting
452: assuming that ozone dry deposition should be a strong function of LAI
455: "complex"
466: "suggesting"
466: suggestion to cut "natural" here and in other spots – natural IAV has ambiguous meaning
475: heterogeneity?
478: soil moisture data?
480: refs for good performance at site level?
495-6: whether IAV in vd at Blodgett is caused by chemistry is unknown
491-497: steps on how authors calculated averages and CVs for long term data needed
499: Olivia has a new paper on this
526: a vague reference to an effort in asia doesn't do much to help the reader
527: "constrain"; why all of a sudden call it gaseous dry deposition?
528: what do the authors mean by reported? do they mean in the peer reviewed literature? there are many reasons why people report fluxes rather than deposition velocities in peer-reviewed publications, and previous work doesn't simply exist to provide deposition velocities for future model evaluation! many datasets are available by contacting the research groups that made them.
536: do the authors actually show that the four parameterizations differ most in leafy parts of the world? if not, i suggest rephrasing
542-544: is this something that is assumed widely?
543: demonstrates that
549: why "at least increase the spatiotemporal representativeness if not the absolute accuracy" – is there some limitation of the Ducker dataset that I am missing?
554-6: the authors could do a better job at illustrating why they are linking these two ideas
561-3: yet the authors barely make use of long-term datasets that are available!
583: what does low baseline vd actually mean?
586: $v_d$
587: do the authors mean the simulation year for the 30% testing?
588: is this somewhat inherently in the LAI product?
593-600: as is, this seems like a stretch to me
608: what is the difference between a model-observation integration and an empirical study?

---

## Referee Comment (RC2) · Anonymous Referee #2 · 23 Jul 2019

Review of
"Importance of Dry Deposition Parameterization Choice in Global Simulations of Surface Ozone" by A. Y. H. Wong, et al.

The paper is an investigation of the impact of different ozone dry deposition parameterizations on the calculation of dry deposition velocities and the global modeling of surface ozone concentrations.  The paper is generally well written and presents a thorough study of the uncertainties generated by the choice of dry deposition parameterization in atmospheric chemistry models.  My only general criticism is that the figures need to be presented in a larger form that will be easier for readers to see. The work is timely and well done and should be published as is with only minor editing changes as suggested below.

Specific Suggestions:

p. 1, line 27:  Should be "The trend in July …", not "trends".

p. 2, line 62:  Should be "… compiled …".

p. 2, line 63:  Should be "measurements from the EC and GM …".

p. 7, line 205:  Should be "… simulation described in the next sub-section."

p. 7, line 211: Should be "… to investigate how …".

p. 7, line 216:  Should be "… the increase in atmospheric …".

p. 8, line 233:  Should be "… developed by NOAA's National Centers for Environmental Prediction (NCEP) and the NASA Global …".

p. 13, line 409:  Should be "… We use the Theil-Sen method …".

p. 14, lines 430-431:  I believe it should be "… a concomitant decrease in July mean surface ozone …".

p. 15, line 461:  Should be "… as they allow for …".

p. 16, line 497:  Should be "… This suggests that the IAV …".

p. 17, line 527:  Should be "… To better constrain regional dry …".

p. 17, line 530:  Should be "… would be greatly benefited if results of ozone flux measurements were reported as both …".

p. 17, line 538:  Should be "… a vast majority of land in …".

p. 18, line 558:  Should be "… mainly concentrates in the drier part of …".

p. 18, line 562:  Should be "… deposition measurements poses …".

p. 18, line 566:  Should be "… The magnitudes of trends are …".

p. 19, line 597:  I believe it should be something like … "… contribute to understanding the role of gaseous dry deposition on air quality, but also to biogeochemical cycling."

p. 19, line 600:  Should be "… global nitrogen cycles."

p. 19, line 607: Should be "… scarcity of measurements."

p. 24, lines 660-661:  The blue dots are very difficult to see on these figures.  The figures should be made larger!

---

## Author Comment (AC1) · 18 Sep 2019

We thank the referee for their positive and constructive comments on our manuscript. We provide our response to each individual reviewer comment (shown in italics) below, including detailed changes to the manuscript (additions in red).

**Major issues:**

1) *The linearity of response of surface ozone concentration to ozone deposition velocity is uncertain, but a major assumption in this study. I'm not convinced that the results from wong et al. 2018 are sufficient to warrant confidence in this assumption. one reason being that they were testing the response to surface ozone to LAI, which involves changes in several processes.*

**Response:**

We agree with the reviewer that our assumption of linearity is important. Our objective with this experiment was to use this first order approach to identify "hotspots" globally where uncertainty/variability in dry deposition velocity could have large impacts on simulated ozone, and then use the assumption of linearity to approximate those impacts. Our approach helps identify regions where more rigorous observations and modeling could be targeted for future work. Still, we address this assumption further. The reviewer notes in particular that the response involves changes in several processes (e.g. non-linearity in chemistry, transport and changes in background ozone).

In response to the reviewer's comment, we have made two changes:

(1) We have changed the manuscript to be more clear about our intentions with using the assumption linearity between perturbations in dry deposition velocity and ozone concentrations, and include a stronger caveat in this interpretation:

 We use this sensitivity to identify areas where local uncertainty and variability in $v_d$ is expected to affect local surface $O_3$ concentration, and we use the assumption of linearity to estimate those impacts to a first order (e.g. Wong et al. 2018). […] However, we note this first-order assumption may not be able to capture the effects of chemical transport, changes in background ozone and non-linearity in chemistry, which can contribute a non-linear response of $O_3$ concentration to $v_d$. Our experiment helps identify regions where more rigorous observation and modeling efforts could be targeted for future work.

(2) To provide an estimate of the error introduced by assuming linearity, we further investigated this assumption in two ways:

      (a) We have mathematically derived an argument for our first-order approximation to calculate $\Delta O_3$ under small $\Delta v_d$, and included this in a new Supplemental Information section.

      (b) We ran another GC sensitivity simulation with 15% increase (instead of the 30% increase) in $v_d$ and to test a second-order approximation to calculate July $\Delta O_3$ with the

Z03_BB deposition parameterization. This approach is compared with our original approach in the new Supplemental Information section.

Based on our analysis, the uncertainty introduced by first-order approximation is within 30%. We have added the following to the manuscript:

In the Methods Section of the manuscript:

We use this sensitivity to identify areas where local uncertainty and variability in vd is expected to affect local surface O3 concentration, and we use the assumption of linearity to estimate those impacts to a first order (e.g. Wong et al. 2018). **In the Supplemental Methods, we justify this first order assumption mathematically, as well as demonstrate the impact of using a second order approximation, and estimate the uncertainty using an assumption of linearity to be within 30%.** However, we note this first-order assumption may not be able to capture the effects of chemical transport, changes in background ozone and non-linearity in chemistry, which can contribute to response of O3 concentration to vd. Our experiment could help identify regions where more rigorous modelling efforts could be targeted in future work.

Supplementary Information, Section 1:

Mathematical analysis for sensitivity of $O_3$ to $\Delta v_d/v_d$:

Assume that $\Delta O_3$ is due to changes in dry deposition flux (with proportionality constant $k_d$) and other first-order processes (e.g. NO titration, loss to $HO_2$ and OH, having total reaction rate $k_c$):

$$dO_3 = d(-k_c O_3 - k_d v_d O_3) \ (S1)$$

Here, $k_c$ and $k_d$ (which are related to meteorology and concentration of other relevant chemical species), are assumed to be relatively constant, so that that the perturbation in $v_d$ does not trigger significant non-linearity. Expanding the differential and rearranging the terms yields:

$$\frac{dO_3}{O_3} = \frac{-k_d \ dv_d}{1 + k_c + k_d} \ (S2)$$

Integrating S2 between perturbed ($O_3 + \Delta O_3$, $v + \Delta v_d$) and unperturbed ($O_3$ and $v_d$) values yields:

$$\ln\left(1 + \frac{\Delta O_3}{O_3}\right) = -\ln\left(1 + \frac{k_d \Delta v_d}{1 + k_c + k_d v_d}\right) \ (S3)$$

Since $\Delta O_3$ is small compared to $O_{3,0}$, first-order expansion is valid. When $\Delta v_d$ is small enough relative to $v_d$ for first-order approximation, Taylor's expansion of S4 yield:

$$\frac{\Delta O_3}{O_3} = -\frac{k_d}{1 + k_c + k_d v_d} \Delta v_d \ (S4)$$

S5 can be rearranged to yield:

$$\Delta O_3 = -\frac{k_d v_d O_3}{1 + k_c + k_d v_d}\frac{\Delta v_d}{v_d} = \beta \frac{\Delta v_d}{v_d}, where \ \beta = -\frac{k_d v_d O_3}{1 + k_c + k_d v_d} < 0 \ (S5)$$

This shows that when the $\Delta v_d/v_d$ is small enough ($\ln(1+x) \approx x$) and does not cause non-linearity ($k_c$ and $k_d$ = constant) in chemistry, $\Delta O_3$ is linearly proportional to $\Delta v_d/v_d$. The error of linearizing the natural logarithms equals to the difference between $\ln(1+x)$ and $x$. This analysis gives the

conditions for when the first-order approximation is reasonable, and allowing us to estimate the error when deviating from these condition. Assuming $\beta$ is correctly estimated by chemical transport model, the error of linearization at $\Delta v_d/v_d = \pm 50\%$ (the upper bound of $\Delta v_d/v_d$ consistent with our analysis), is on the order of 25%. For more typical value of $\Delta v_d/v_d$ (20%), the error is around 10%.

As $\Delta v_d/v_d$ gets larger, we can expand R.H.S of S3 to the second order and investigate sensitivity of $\Delta O_3$ to $\Delta v_d/v_d$:

$$\Delta O_3 = \beta \frac{\Delta v_d}{v_d} - \frac{\beta^2}{2O_3}\left(\frac{\Delta v_d}{v_d}\right)^2 = \left(\beta - \frac{\beta^2}{2O_3}\frac{\Delta v_d}{v_d}\right)\left(\frac{\Delta v_d}{v_d}\right) = \beta' \frac{\Delta v_d}{v_d} \quad (S6)$$

Where $\beta'$ is the "corrected $\beta$", which is a function of $\Delta v_d/v_d$.

To illustrate the potential impact of such non-linearity on $\Delta O_3$, we compare July $\Delta O_{3,Z03\_BB}$ estimated using first-order estimation with $\beta$ derived from $\Delta v_d/v_d = +15\%$ and $+30\%$, and second-order approximation, and the result is shown in figure S1. The three different methods produce very similar $\Delta O_3$, and their differences have little impact on our conclusion. For simplicity, we only show the result using $\beta$ derived from $\Delta v_d/v_d = +30\%$ in the main manuscript.

As noted above and in the main manuscript, our approach is limited by the assumption that chemistry and transport do not introduce non-linear terms which may not be realistic. Rather, our approach is intended to identify hotspots of impact, and quantify these potential impacts to a first order. More rigorous modeling efforts could then be targeted in future work.

Supplemental figure 1:

[Figure]

Figure S1. July $\Delta O_{3,Z03\_BB}$ calculated using a) first-order method where $\beta$ is derived from $\Delta v_d/v_d = +30\%$ GC sensitivity run, b) first-order method where $\beta$ is derived from $\Delta v_d/v_d = +15\%$ GC sensitivity run, and c) second-order method with $\beta$ derived from $\Delta v_d/v_d = +15\%$.

2) *The authors' attribution of biases and intermodel differences are entirely speculative. there is no rigorous evaluation of the processes/aspects leading to differences. I tend to not be in favor of such speculation and I think it masks the strength of the model evaluation (that not any one parameterization is best or worst) and model intercomparison.*

**Response:**

We appreciate the reviewer's caution, and do not want to detract from other strengths of the manuscript. We have identified several speculative statements in our model evaluation, and have removed them from the manuscript.

We have removed the following statements from the manuscript:

……

……

……

We believe additional cases are addressed in response to the Reviewer's minor comments below.

**Minor issues:**

*10: I tend to think the sinks of ozone are chemistry and dry deposition so "second largest sink" doesn't say much to me:*

Response: We agree with the reviewer that this wording is unnecessary. In response, we have changed our manuscript to:

**Dry deposition is  a major sink of tropospheric ozone.**

*15-16: "to drive four ozone dry deposition parameterizations"*

Response: We have made the suggested changes:

**We use consistent assimilated meteorology and satellite-derived leaf area index (LAI) to drive four ozone dry deposition parametrizations  that are representative of the current approaches of global ozone dry deposition modelling over 1982-2011 …**

*62: I wouldn't say Silva & Heald 2018 is a review*

Response: We have made the suggested correction:

**A recent  study (Silva and Heald, 2018)…**

*66: "account for" is vague; in general this sentence implies canopy column models are better than big-leaf ones, which has yet to be shown in the literature,* and

*67: the authors said previously that reaction with BVOCs is a nonstomatal pathway so here saying that it is in addition to surface sinks is a little confusing*

> Response: We agree that "account for" is vague. Our intention was to discuss the additional processes and details that canopy column model simulates comparing to big-leaf model, rather than commenting which one is better (in terms of more accurate simulation of $v_d$). In response to the reviewer comment we have reworded this:
>
> **…which are able to  simulate the effects vertical gradients inside the canopy environment, and gas-phase reaction with BVOCs …**

*67-71: canopy column models still use resistance networks …*

> Response: We agree that the main difference between canopy column model and general CTM parameterizations is multi-layer vs big-leaf representation, rather than the use of resistance network. In response to the reviewer comment we have reworded this.
>
> **…and horizontal resolution for resolving the plant canopy in such detail, instead represent plant canopy foliage as 1 to 2 big leaves, and  $v_d$ is parameterized as a network of resistance…**

*77-80: this has yet to be shown… these formulations can be variable across models …*

> Response: We acknowledge the formulations can be variable across model. Wu et al. (2018) show that out of the 4 big-leaf parameterizations that are considered in their work, all of them shares very similar formulae for $r_b$. $r_a$ is mostly based using Monin-Obukhov similarity theory and the difference in universal function is not found to affect $v_d$ significantly. Other parameterizations that are not included in that study (e.g. Simpson et al., 2012) often use very similar formulae for $r_b$ and Monin-Obukhov similarity theory for $r_a$. In response to the reviewer comment, we have reworded this:
>
> **The calculation of $R_a$ (mostly based on Monin-Obukhov similarity theory) and $R_b$…**

*80-88: the connection between these paragraphs (last sentence of previous one and first sentence of next one) could be articulated better*

> Response: We agree with this suggestion. In response, we have added the following wording:
>
> **Such formalism is empirical in nature and does not adequately represent the underlying ecophyioslogical processes affecting $R_s$ (e.g. temperature acclimation). An advance of these efforts includes harmonizing $R_s$ with that computed by land surface models…**

*101: Hardacre et al. show factor of 2-3 differences across models - so are all models' seasonal cycles well represented? also I suggest changing "demonstrating" to "suggesting".*

> Response: We agree with the reviewer. In response to the reviewer comment, we have reworded this sentence:
>
> **This work    suggests that the difference in land cover classification is the main source of discrepancy between models…**

*125: "unable" seems harsh; it doesn't seem Clifton et al. even tried to do this*

> Response: We have reworded this:
>
> **...although they  do not show how the IAV of $v_d$ may contribute to the IAV of $O_3$…**

*128: cut "physics"*

> Response: We have made this change as requested.

*145: I find the placement/existence of this sentence strange. the authors don't investigate the same parameterizations that Wu et al. do.*

> Response: We agree with the reviewer, and have removed this sentence.

*153: refs for strong empirical relationship*

> Response: In response to the reviewer comment, we have added references to this statement:
>
> **…strong empirical relationship between photosynthesis ($A_n$) and stomatal conductance ($g_s$) (e.g. Ball et al., 1987; Lin et al., 2015)…**

*162-173: I see that the authors have basically organized their parameterizations according to model (w/ exception of #2)*

*1) The GEOS Chem parameterization*

*2) Zhang parameterization*

*3) The CESM parameterization*

*4) The UKCA parameterization*

*I didn't realize this at first and the parameterizations chosen seemed quite strange. I would urge the authors to re-frame their parameterization description (but also noting that their parameterizations are not exact replicates of a given model)*

Response: We agree with the reviewer that our choice of configurations is broadly implemented in some CTMs as mentioned. We intentionally separated our choice of parameterizations from their actual implementation in CTMs because we want our result to be representative of classes of approaches of modelling $v_d$, as we have explained this in line 150 – 158. Furthermore, the choice of doing Z03_BB and W98_BB comes from recent efforts to harmonize CTM $R_s$ with Earth System Model/Land Surface Model $R_s$ as a viable option for improving $v_d$ simulations (line 86). However, we agree with the reviewer that we could reframe these descriptions to be more clear, and use examples in their description. In response to the reviewer's comment we have made the following changes:

**3) W89 with *Rs* calculated from a widely-used coupled *An-gs* model, the Ball-Berry model (hereafter referred to as W98_BB*) (Ball et al., 1987; Collatz et al., 1992, 1991), which is similar to that proposed by Val Martin et al. (2014), and therefore the current parameterization in Community Earth System Model (CESM). This represents Type 3 in stomatal and Type 1 in non-stomatal parametrization.**
**4) Z03 with the Ball-Berry model (Z03_BB), which is comparable to the configuration in Centoni (2017) implemented in United Kingdom Chemistry and Aerosol (UKCA) model. This represents Type 3 in stomatal and Type 2 in non-stomatal parametrization.**

*175: It doesn't quite make sense to me that the authors say the Zhang parameterization is "open source" in one sentence and a couple sentences later say that implementing it required personal communication with Zhiyong and Leiming.*

Response: This is a good point. We deleted the word "open-source".

*180: Given that GEOS Chem doesnt have a land surface model, I think the authors need to clarify how exactly Anet is calculated.*

Response: We have added a brief description of the $A_n$-$g_s$ model in the new supplemental material section:

A brief description of photosynthesis-stomatal conductance ($A_n$-$g_s$) module in TEMIR (a manuscript is in prep)

TEMIR largely follows Oleson et al. (2013), where net photosynthetic rate ($A_n$, μmol $CO_2$ m$^{-2}$ s$^{-1}$), stomatal conductance for water ($g_{sw}$, μmol m$^{-2}$ s$^{-1}$) and $CO_2$ concentration in leaf mesophyll ($c_i$, mol mol$^{-1}$) are solved simultaneously by the following coupled set of equations:

$$A_n = \frac{g_{sw}}{1.6}(c_a - c_i) \ (S7)$$

$$g_{sw} = \beta_t g_0 + g_1 \frac{A_n}{c_s} RH_s \ (S8)$$

$$A_n = A - R_d \ (S9)$$

Here, $c_a$ is $CO_2$ concentration (mol mol$^{-1}$), $\beta_t$ is soil moisture stress factor (unitless), $g_0$ is minimum stomatal conductance (μmol m$^{-2}$ s$^{-1}$), $A_n$ is net photosynthetic rate (μmol $CO_2$ m$^{-2}$ s$^{-1}$), $A$

is gross photosynthetic rate (μmol $CO_2$ m$^{-2}$ s$^{-1}$) and $R_d$ is dark respiration rate (μmol $CO_2$ m$^{-2}$ s$^{-1}$). Furthermore, $c_s$ and $RH_s$ are the $CO_2$ concentration (mol mol$^{-1}$) and relative humidity (unitless) at leaf surface. $A$ is calculated following Bonan et al. (2011), which is based on Farquhar et al. (1980) and Collatz et al. (1992):

$$\Theta_{cj}A_i^2 - (A_c + A_j)A_i + A_cA_j = 0 \quad (S10)$$

$$\Theta_{ip}A^2 - (A_i + A_p)A + A_iA_p = 0 \quad (S11)$$

For C3 plants, $\Theta_{cj} = 0.98$ and $\Theta_{ip} = 0.95$. For C4 plants, $\Theta_{cj} = 0.80$ and $\Theta_{ip} = 0.95$. Rubisco-limited rate ($A_c$, μmol $CO_2$ m$^{-2}$ s$^{-1}$), light-limited rate ($A_j$, μmol $CO_2$ m$^{-2}$ s$^{-1}$), product-limited rate ($A_p$, μmol $CO_2$ m$^{-2}$ s$^{-1}$) and $R_d$ are calculated as:

$$A_c = \begin{cases} \dfrac{V_{c\,max}(c_i - \Gamma_*)}{c_i + K_c(1 + \dfrac{0.21P_{atm}}{K_o})} & for\ C_3\ plants \\ V_{c\,max} & for\ C_4\ plants \end{cases} \quad (S12)$$

$$A_j = \begin{cases} \dfrac{J(c_i - \Gamma_*)}{4c_i + 8\Gamma_*} & for\ C_3\ plants \\ 0.23\phi & for\ C_4\ plants \end{cases} \quad (S13)$$

$$A_c = \begin{cases} 3T_p & for\ C_3\ plants \\ k_p\dfrac{c_i}{P_{atm}} & for\ C_4\ plants \end{cases} \quad (S14)$$

$$R_d = \begin{cases} 0.015V_{c\,max} & for\ C_3\ plants \\ 0.025V_{c\,max} & for\ C_4\ plants \end{cases} \quad (S15)$$

Here, $V_{cmax}$, $\Gamma_*$, $P_{atm}$, $J$, $\varphi$, $T_p$ and $k_p$ are the maximum rate of carboxylation (μmol m$^{-2}$ s$^{-1}$), $CO_2$ compensation point (mol mol$^{-1}$), atmospheric pressure (Pa), electron transport rate (μmol m$^{-2}$ s$^{-1}$), absorbed photosynthetically active radiation (PAR) (W m$^{-2}$), triose phosphate utilization rate (μmol m$^{-2}$ s$^{-1}$) and initial slope of C4 $CO_2$ response curve (μmol Pa$^{-1}$ m$^{-2}$ s$^{-1}$). $K_c$ and $K_o$ are the Michaelis-Menten constants for $CO_2$ and $O_2$ (Pa). Furthermore, $J$ is calculated as the smaller root of the following equation:

$$0.7J^2 + (1.955\phi + J_{max})J + 1.955\phi = 0 \quad (S16)$$

Where $J_{max}$ is the maximum potential rate of electron transport (μmol m$^{-2}$ s$^{-1}$). As $J_{max}$, $\varphi$, $V_{cmax}$ and the variables related to $V_{cmax}$ ($\Gamma_*$, $J_{max}$, $T_p$, $R_d$) differ between sunlit and shaded leaves, the above set of equations are solved separately for sunlit and shaded leaves.

The parameters ($V_{cmax}$, $\Gamma_*$, $K_c$, $K_o$, $J_{max}$, $T_p$, $R_d$) are functions of vegetation temperature ($T_v$), and the temperature scaling formulae are given at eq. 8.9 to eq. 8.14, while the effect of temperature acclimation (Kattge and Knorr, 2007) on $J_{max}$ and $V_{cmax}$ are given at eq. 8.15 and 8.16 in Oleson et al. (2013). Other details of the model formalism (e.g. canopy scaling and effect of $\beta_t$ on $V_{cmax}$) also follow Chapter 8 in Oleson et al. (2013), therefore we will focus on describing the main differences between CLM 4.5 and TEMIR.

First, TEMIR is driven entirely by assimilated meteorology. Instead of solving the whole surface energy balance equation, TEMIR consistently calculates $T_v$ from 2-meter air temperature ($T_2$, K)

and sensible heat flux ($H$, W m$^{-2}$) using Monin-Obukhov similarity theory (Monin and Obukhov, 1954):

$$T_v = T_2 + \frac{H}{\rho c_p}(r_{a,h} + r_{b,h}) \ (S16)$$

Where $\rho$, $c_p$, $r_{a,h}$ and $r_{b,h}$ are air density (kg m$^{-3}$), specific heat of air at constant pressure (J kg$^{-1}$ K$^{-1}$), aerodynamic and laminar boundary-layer resistance (s m$^{-1}$) of heat, respectively.

Secondly, MERRA-2 only provides soil moisture output at two levels (surface and root zone), which is not compatible with the multi-layer soil module in CLM. Therefore, instead of aggregating $\beta_t$ from multiple soil layers, TEMIR calculates $\beta_t$ from the root-zone soil wetness of MERRA-2. Soil wetness ($s$) is first converted into soil matric potential ($\psi$, mm) using the following equation:

$$\psi = \psi_{sat} s^{-B} \ (S17)$$

Where $\psi_{sat}$ and $B$ are the soil matric potential (mm) at saturation and Clapp-Hornberger exponent (Clapp and Hornberger, 1978), which are related to soil property. Then $\beta_t$ is calculated as:

$$\beta_t = \frac{\psi_c - \psi}{\psi_c - \psi_0}\left(\frac{\theta_{sat} - \theta_{ice}}{\theta_{sat}}\right), 0 \le \beta_t \le 1 \ (S18)$$

Where $\psi_c$ and $\psi_0$ are the soil matric potential (mm) at which stomata are full close or fully open, and the term in the bracket account for the fact that frozen water are not available for plants.

*182-183: It's fine not to test Ra and Rb, but i suggest that the authors do not use this qualifier. This isn't well understood (Does Fares et al. even show this?)*

Response: In response to the reviewer's comment we have deleted this qualifier:

**…which is numerically stable (Sun et al., 2012). …**

*188-9: has this model been evaluated? or used previously?*

Response: An evaluation paper of this model is in prep by collaborators who hope to have this submitted shortly in a Discussion format, and we intend to add this citation if possible before publicatio.

*194-5: what are these variables used for?*

Response: These variables are needed to drive the dry deposition parameterizations, as they require land cover classification (basically PFT) and LAI. Soil property is required for running the $A_n$-$g_s$ model. In response to the reviewer's question we have added the following text to our manuscript:

**…We use the CLM land surface dataset (Lawrence and Chase, 2007), which contains information for land cover, per-grid cell coverage of each plant functional type (PFT) and PFT-specific LAI, which are required to drive the dry deposition parametrizations, and soil property, which is required to drive the $A_n$-$g_s$ model in addition to PFT and PFT-specific LAI.**

*195: presumably the authors' decisions about land type mapping (& differences for "W89" vs "Z03") impact the authors' results… one implication of this is that the authors' statement in the abstract or introduction that the only thing different across parameterizations is the model structure is not necessarily true*

Response: The reviewer raises an excellent point. We agree that this is one of the key uncertainty of our approach and deserves more discussion. This is mostly limited to herbaceous and shrub land type as the CLM forest PFT correspond pretty well to W98/Z03 land types. In response to the reviewer's comment, we added the following:

**… do not resolve croplands into such detail. Having land cover maps that distinguish between more crop types could potentially improve the performance of Z03. The evaluation for herbaceous land types also suggests that as CLM PFT do not have exact correspondence with W98 and Z03 land types, our results over herbaceous land types are subject uncertainty in land type mapping (e.g. tundra vs grassland, specific vs generic crops, C3 vs C4 grass).**

*197: I would suggest cutting the "(eg. leaf physiological and soil hydrauilic constants)" - becoming more specific here doesn't help readers when the parameterizations are not laid out and we have no idea what these terms do/stand for*

Response: This was removed as suggested.

*198: what's z0?*

Response: We have clarified this in the manuscript:

**… while land-cover specific roughness length ($z_0$) values follow Geddes et al. (2016).**

*198: how is leaf wetness calculated? how is snow calculated?*

Response: We have added the following to our manuscript:

**…follow Geddes et al. (2016). Leaf is set to be wet when either latent heat flux < 0 W m$^{-2}$ or precipitation > 0.2 mm hr$^{-1}$. Fractional coverage of snow for Z03 is parameterized as a land-type specific function of snow depth following the original manuscript of Z03, while W98 flags grid cells with albedo > 0.4 or permanently glaciated as snow-covered.**

*203: how do the authors scale PFT-specific LAI? is there an established method of doing this? presumably this has implications for the findings*

Response: We choose to derive scaling factors as the direct disaggregation method of Lawrence and Chase (2007) is very difficult to replicate, and derived the grid-cell level scaling factor at $2° \times 2.5°$ by comparing the monthly mean LAI at each year with that of the 30-year mean. In theory PFT-specific LAI can be simulated by land surface model, but that will be even more uncertain and less empirically-constrained then using satellite LAI. In response to the reviewer comment, we clarify our approach in the manuscript:

**We  derive the interannual scaling factors as the ratio between the monthly LAI at specific year and that of the 30-year mean of GIMMS LAI3g, that can be applied to scale the baseline CLM-derived LAI (Lawrence and Chase, 2007) for each month over 1982 to 2011…**

*217: I think the authors need to articulate here or in the introduction the various effects that high CO2 may have on ozone dry deposition velocity and the various uncertainties in our understanding of CO2 fertilization (& reference previous work examining this)*

Response: We added the following:

**…enhanced cuticular O3 uptake under leaf surface wetness (Altimir et al., 2006; Potier et al., 2015, 2017; Sun et al., 2016). Furthermore, terrestrial atmosphere-biosphere exchange is also directly affected by $CO_2$, as $CO_2$ can drive increases in LAI (Zhu et al., 2016) while inhibiting $g_s$ (Ainsworth and Rogers, 2007). These can have important implications on $v_d$, as shown by Sanderson et al. (2007), where doubling current $CO_2$ level reduces $g_s$ by $0.5 - 2.0$ mm s$^{-1}$, and by Wu et al. (2012) where $v_d$ increases substantially due to $CO_2$ fertilization at 2100. Observations from the Free Air $CO_2$ Enrichment (FACE) experiments also $CO_2$ fertilization and inhibition of $g_s$ effects, but the impacts are variable and species specific such that extrapolation of these effects to global forest cover is cautioned (Norby and Zak, 2011).**

*229: is the proper/up-to-date way of referencing GEOS-Chem?*

Response: We have replaced the citation to Bey et al. (2001) with a link to the GEOS-Chem model, which is up-to-date and we believe is consistent with the GEOS-Chem community's approach (in addition to including citations to the most relevant developments in the GEOS-Chem chemistry, as we have done).

*237: binned = jargon*

Response: We changed the sentence to:

**Both of the maps are  remapped from their native resolutions to $0.25° \times 0.25°$.**

*243-246: discussing about dry deposition of other species and impacts on ozone requires introducing some concepts (or cutting talking about dry deposition of other species)*

Response: We removed the sentence talking about dry deposition of other species.

*249-251: this seems like a strange choice to me. it's not differences in transport per se, it's differences in background ozone caused by changes in ozone dry deposition. why wouldn't the authors want to capture this? because it contributes to nonlinear responses to ozone dry deposition?*

Response: We agree with reviewer's comment that perturbation in $v_d$ causes changes in background $O_3$, and this can be potentially important. Our main objective is to study the local uncertainty in $O_3$ due to local uncertainty in $v_d$. Therefore, we choose to limit our study to regions with sufficiently high $v_d$, where the changes and uncertainties in surface $O_3$ are more likely to be dominated by the direct effect of $v_d$ rather than changes in background $O_3$, and avoid the potential non-linearity as suggested by the reviewer. In response to the reviewer's comment, we have clarified this in our mauscript:

 **We use this sensitivity to identify areas where local uncertainty and variability in $v_d$ is expected to affect local surface $O_3$ concentration, and we use the assumption of linearity to estimate those impacts to a first order (e.g. Wong et al. 2018).**

**…are expected to be attributed more to**  **changes in background $O_3$ rather than…**

*249: what is the baseline simulation?*

Response: We make the following change:

**…where the monthly average $v_d$ is greater than 0.25 cm s$^{-1}$ in the**  **unperturbed GEOS-Chem simulation…**

*254: Why not CLIM+LAI+CO2 as well?*

Response: As we show later, over these 30 years, $CO_2$ has very minor effect on $v_d$ (fig. 9). In response to the reviewer's question, we have added the following:

**…largely based on the evaluation presented in Silva and Heald (2018). We do not include the evaluation of $v_d$ from [Clim+LAI+CO$_2$] scenario as we find that the impact of $CO_2$ concentration on $v_d$ is negligible over the period of concern, as we will show in subsequent sections.**

*261-3: How many sites does this cut?*

Response: This removes 1/3 of the original data (25 data sets). In response to the reviewer's question, we have added this:

**While this**  **removes 1/3 of the original data sets used in Silva and Heald (2018)…**

*265: Fractional coverage of what? (please spell out in text) Why are these figures shown? they are not very useful for the reader*

Response: For fractional coverage, we refer to "each major land type" in line 267. We do agree that our description did not help readers to understand the graph. In response to the reviewer questions, we have made the following changes:

**Nearly all the observations are clustered in Europe and North America, except three sites in the tropical rainforest and one site in tropical deciduous forest in Thailand. For most major land types, there are significant mismatches between the locations of flux measurements and the dominant land cover fraction, which may hinder the spatial representativeness of our evaluation.**

Response: We agree that the statement is unnecessary. We have removed the sentence.

*273: it seems strange to me that the authors would generalize such as bias, given that it's unclear if the bias is caused by a particular attribute of a land type or process, and that the land type-specific biases differ across the parameterizations*

Response: We agree with the reviewer that it may not be a good choice to generalize such bias. We have made the following change:

 **The performance metrics of each parameterization at each land type are summarized in table 2.**

Response: Thanks for pointing out this ambiguity. We have made the following changes:

**… by the four dry deposition parameterizations, with *N* referring to number of data points (1 data point = 1 seasonal mean).**

*288: if the authors are implying ambient chemistry is happening then they should just say it*

Response: As suggested in earlier response, this sentence contains unnecessary speculation and therefore we have deleted the sentence.

Response: Yes. We have clarified this in our manuscript:

**…as most global land cover data sets do not resolve croplands into such detail. Having land cover maps that distinguish between more crop types could potentially improve the performance of Z03…**

*301-302: I'm not sure that the following lines illustrate this; in other words, i think BB "generally but not universally leads to improvements" is not supported by the actual findings — it seems to be for Z03 — but not for Wesely — which may suggest that we need to be paying attention to nonstomatal deposition estimates too.*

Response: We agree that non-stomatal deposition should not be overlooked, and we agree that the improvement of Z03_BB over Z03 is more significant than that of W98_BB over W98. We find that W98_BB and W98 have comparable performance over forests, but W98_BB significantly outperform W98 over herbaceous land types. We also agree that nonstomatal parameterization probably contributes to the different responses between W98_BB vs W98 and Z03_BB vs Z03. We changed our wording as follows:

**…improving spatial distribution of mean $v_d$. The different responses to substituting native $g_s$ with that from Ball-Berry model highlight the significant differences in parameterizing non-stomatal uptake between W98 and Z03, which further suggests that the uncertainty in non-stomatal deposition should not be overlooked.**

*313-4: what particular problem has been highlighted?*

Response: This refers to the mismatch between EC footprint and model resolution. In response to the reviewer comment, we have clarified this in our manuscript:

** The mismatch between model resolution and the footprint of site-level measurements…**

*315: sampling biases meaning that the authors are not evaluating most locations on earth, right? the authors are sampling the time/place of the measurements*

Response: This is correct and we acknowledge our ambiguity in wording. We make the following change:

**… the evaluation may be further compromised by inherent spatial sampling biases (fig. 1).**

*317-320: not sure what the point of this paragraph is. what is the hypothesis being investigated?*

Response: The main purpose of the section is to show that our model implementation gives reasonable result at seasonal scale. Comparing the W98 result from our implementation with that from GC further supports our argument. In response to the reviewer question, we have added the following wording:

**W98 run with static LAI, providing further evidence that our implementation of W98 is reliable…**

*334-5: recommend that the authors don't speculate here or elsewhere*

Response: We agree that it is unnecessary. We removed the speculative statement:

**In India, Australia, western US, and polar tundra Mediterranean region, July mean daytime $v_d$ is low (0.2 - 0.5 cm s$^{-1}$). **

*349-50: on a similar note as the above comment, how do the authors know this?*

Response: We agree that we should provide more information to support our argument, and will make our explanation much clearer. We added the soil moisture stress factor map as figure S2. In July, over southern Africa, the soil stress factor is exceptionally low, indicating that drought stress does strongly limit $g_s$ over the region. We have also changed our wording to be more cautious in our interpretation (instead of "because", we state, "which may be due to". We changed line 350 to:

**…which may be due to the explicit consideration of soil moisture limitation to $A_n$ and $g_s$ (demonstrated by the spatial overlap with soil moisture stress factors shown in Fig. S2)…**

And in the Supplemental Information:

[Figure]

**Figure S2. July average soil moisture stress factor ($\beta_t$). $\beta_t = 1$ represents no soil moisture stress, while smaller $\beta_t$ means stronger soil moisture stress and more stomatal closure. $\beta_t = 0$ signifies that soil moisture stress is so strong that it completely shuts down stomatal activity.**

*353: "is not desiccated"?*

Response: We have clarified this in our manuscript as follows:

**… as long as the soil does not  become too dry to support stomatal opening…**

*358: i don't think the authors show this; they just speculate that this is the cause.*

Response: In response to the reviewer's comment, we have omitted this comment.

*368: will the authors more carefully articulate what Centoni finds so that the reader knows how to compare the findings*

Response: We agree this reference may not be ideal since Centoni (2017) did not explicitly talk about all four parameterizations. We have removed this reference:

**…We find ΔO₃ is the largest in tropical rainforests for all the parameterizations (up to 5 to 8 ppbv) …**

*370: i assume that the authors are identifying the hot spot regions through their large delta O3. related: perhaps the authors are missing a delta on the v_d,i in Equation 3.*

Response: We thank the reviewer for catching this oversight. We have amended equation (3) to:

$$\Delta O_3 \approx \beta \frac{\Delta \overline{v_{d,i}}}{\overline{v_{d_{W98}}}} \ (3)$$

*378: are the authors really "exploring the importance of seasonality in predictions of vd and their subsequent impact" with their current approach? (see comment below for line 404)*

*404: are the authors actually showing the impacts on seasonality? showing the impact in each season is not the same as showing the impact on seasonality (a couple of easy calculations could help here)*

Response: We agree with the reviewer that "showing the impacts in different season" is not equivalent to "showing the impact on seasonality". In response to the reviewers' comments, we have clarified our intentions:

**To explore the impact of different prediction of $v_d$ on surface O₃ in different seasons, …**

**…not only affects the mean of predicted surface ozone, but also has different impacts in different seasons...**

*382-4: i suggest a semi colon connecting these two sentences*

Response: Changed as suggested

*385: "shifts from the south to the north relative to July"*

Response: Changed as suggested

*387: i'm not a fan of the authors' use of the term hydroclimate — it's vague — can the authors just say soil moisture or VPD or leaf wetness?*

Response: We agree that "hydroclimate" is vague. We have clarified this in the manuscript:

**… driven primarily by the response to hydroclimate-related parameters such as soil moisture, VPD and leaf wetness, in addition to  land type-specific parameters…**

*398: the suggestion that "hydroclimate [is] a key driver of process uncertainty" seems limited to the tropics/subtropics. am i correct in this interpretation? if so, this should be emphasized.*

Response: We agree with this interpretation. We have clarified our interpretation as follow:

**These findings identify hydroclimate as a key driver of process uncertainty of vd over tropics and subtropics, and therefore its impact on the spatial distribution of surface ozone concentrations, independent of land type-based biases, in these regions.**

*409: briefly describe this method such as the limitations/strengths of it*

Response: We add this to our manuscript:

**We use Theil-Sen method (Sen, 1968), which is less susceptible to outliers than least-square methods, to estimate trends…**

*413: what trends? trends in meteorology, LAI, and/or CO2?*

Response: We are referring to $v_d$. We have clarified this:

**Figure 9 shows the potential impact of  the trends in $v_d$ on…**

*415: how is the annual change in vd estimated? is it using the Theil-Sen method? this part needs better explanation; the reader needs to at least have some concept of what the method used is*

Response: We have clarified our methods as follows:

$$\Delta O_{30y,i} \approx \beta \times m_{v_{d,i}} \times 30 \ (4)$$

**where $\Delta O_{3\ 30y,i}$ and $m_{vd,i}$  are the absolute change in ozone inferred to a first order as a result of the trend of $v_d$ and the normalized Theil-Sen slope (% yr$^{-1}$) of $v_d$, for parameterization $i$ the over the 30-years (1982-2011).**

*423-4: but they are small or nonsignificant per the first line of the paragraph?*

Response: Our wording here was indeed confusing. We have clarified this by making the following changes:

**In [Clim] simulations (where LAI is held constant),  significant decreasing trends in July daytime vd are simulated by the Z03, W98_BB and Z03_BB  over Mongolia, where significant increasing trend in T (warming) and decreasing trend in RH (drying) detected in the MERRA-2 surface meteorological field in July daytime **

*439: or it may decrease as plants acclimate or as nutrients become limiting*

We acknowledge that the sensitivity of terrestrial biosphere to CO2 can be highly variable. In response to the reviewer's comment, we have elaborated and included citations to related literature:

**We note that the importance of the CO2 effect could grow as period of study further extend to allow larger range of atmospheric CO2 concentration (Hollaway et al., 2017; Sanderson et al., 2007). **

*452: assuming that ozone dry deposition should be a strong function of LAI*

Response: We have clarified this statement to include this correction:

**…since both stomatal and non-stomatal conductance in W98 are assumed to be strong functions of LAI…**

*455: "complex"*

Response: Changed as advised

*466: "suggesting"*

Response: Changed as advised

*466: suggestion to cut "natural" here and in other spots - natural IAV has ambiguous meaning*

Response: Cut as advised

*475: heterogeneity?*

Response: We changed this line:

**… show more spatial  heterogeneity compared to W98 and Z03.**

*478: soil moisture data?*

The advent of microwave remote sensing data provides excited opportunities for assimilating soil moisture. However, converting soil moisture into soil matric potential, which is measures of attraction between soil matrix and water, and therefore ecophysiologically relevant, requires data of soil property, which is less constrained globally. In response to this comment, we have made the following clarification:

**Given the uncertainty in  global soil property maps (Dai et al., 2019)…**

*480: refs for good performance at site level?*

Response: We have added a reference as follows:

**… despite their relatively good performance in site-level evaluation (e.g. Wu et al., 2011).**

*495-6: whether IAV in vd at Blodgett is caused by chemistry is unknown*

Response: We agree that our wording is ambiguous. Rather than claiming chemistry causing IAV, we have reworded this sentence as follows:

**In Blodgett Forest, **

*491-497: steps on how authors calculated averages and CVs for long term data needed*

Response: We agree with the reviewer that this subsection would benefit from some clarification. As most of the IAV section presents result for July, we now recalculate and present the July $CV_{vd}$ for all the 3 sites based on the raw data, and the details of calculation is given in supplemental material. We calculated July mean daytime $v_d$ for each year by averaging the individual hourly averages to avoid hourly sampling bias, and derive $CV_{vd}$ by dividing the standard deviation by the mean of July mean $v_d$ over all years. The recalculated numbers do not change our conclusion significantly. In response to the reviewer's suggestion, we have made the following changes to our manuscript:

**We compare the simulated  July $CV_{vd}$ from all four deposition parameterizations with those recorded by publicly available long-term observations. Hourly $v_d$ is calculated using eq. (1) from raw data. We filter out the data points with extreme (> 2 cm s⁻¹) or negative $v_d$, and without enough turbulence ($u_* < 0.25$ m s⁻¹). As $v_d$ in each daytime hours are not uniformly sampled in the observational datasets, we calculate the mean diurnal cycle, and then calculate the daytime average July of $v_d$ for each year from the mean diurnal cycle, from which $CV_{vd}$ can be calculated.**

**The IAV predicted by all four parameterizations at Harvard Forest is between 3% to 7.9%, which is 2 to 6 times lower than that presented in the observations ( 18%).  We find similar underestimates by all four parameterizations compared to the long-term observation from Hyytiala (Junninen et al., 2009; Keronen et al., 2003; https://avaa.tdata.fi/web/smart/smear/download), where observed $CV_{vd}$ (16%) is significantly higher than that predicted by the deposition parameterizations (3.5% - 7.1%). In Blodgett Forest, where $O_3$ uptake is more  attributable to gas-phase reactions (Fares et al., 2010; Wolfe et al., 2011), we find that the models underestimate the observed annual $CV_{vd}$ more seriously (~1%– 3% compared to  18% in the observations)**

*499: Olivia has a new paper on this*

Response: We agree that Olivia's new paper is an excellent reference of furthering our discussion. We have added this reference:

**Clifton et al. (2019) attribute this to the IAV in deposition to wet soil and dew-wet leaves, and in-canopy chemistry under stressed condition for forests over northeastern U.S.  Some of these processes (e.g. in-canopy chemistry, wetness slowing soil ozone uptake) are not represented by existing parameterizations, contributing to their  difficulty in reproducing the observed IAV .**

*526: a vague reference to an effort in asia doesn't do much to help the reader*

Response: We add the reference to the measurement in Asia as follow:

**We know of only one multi-season direct observational record in Asia (Matsuda et al., 2005) and none in Africa…**

*527: "constrain"; why all of a sudden call it gaseous dry deposition?*

Response: We agree that our paper does not discuss about other gaseous species. We clarified this:

**To better constraint regional $O_3$ dry deposition, effort must be made in making new observations …**

*528: what do the authors mean by reported? do they mean in the peer reviewed literature? there are many reasons why people report fluxes rather than deposition velocities in peer-reviewed publications, and previous work doesn't simply exist to provide deposition velocities for future model evaluation! many datasets are available by contacting the research groups that made them.*

Response: We agree that our wording could be misinterpreted and requires clarification. We have simplified the text in our manuscript:

~~We also find that many existing ozone flux measurements are not usable for our evaluation purposes, since only FO3 is reported in detail instead of vd. Evaluation and development of ozone dry deposition parameterizations would be greatly benefited if result of ozone flux measurements is reported in both FO3 and vd, or even have publically available ozone flux and other related micrometeorological variables, which allows both direct evaluation of vd and solves the mismatch between coarse model grids and the site (e.g. Wu et al., 2011, 2018).~~ **Evaluation and development of ozone dry deposition parameterizations will continue to benefit from publicly available ozone flux measurements and related micrometeorological variables that allow for partitioning measured flux into individual deposition pathways (e.g. Clifton et al., 2017; 2019, Fares et al., 2010, Wu et al., 2018****).**

*536: do the authors actually show that the four parameterizations differ most in leafy parts of the world? if not, i suggest rephrasing*

Response: We have rephrased this statement.

**We find that these discrepancies are in general a function of both location and season. In NH summer, $v_d$ simulated by the 4 parameterizations are considerably different in many**  **regions over the world.**

*542-544: is this something that is assumed widely?*

Response: We have reworded this statement:

**This demonstrates the potential impact of parameterization choice (or, process uncertainty) on $v_d$ is neither spatiotemporally uniform nor negligible in**  **many** **regions over the world.**

*543: demonstrates that*

Response: Changed as advised

*549: why "at least increase the spatiotemporal representativeness if not the absolute accuracy" - is there some limitation of the Ducker dataset that I am missing?*

Response: This is because FLUXNET-based data can provide a constraint stomatal deposition, but with limited opportunity to constrain other individual pathways. Potentially, if the biases in stomatal and non-stomatal deposition offsets each other, constraining stomatal deposition may lead to substantial biases in $v_d$. Whether better constrained $g_s$ leads to significantly better constrained $v_d$ we believe is still an open research question and is something we are investigating in a follow up study. In response to the reviewer's question, we have added the following text to the manuscript:

**…increase the spatiotemporal representativeness, if not the absolute accuracy, of dry deposition parameterization,** **since it would be difficult to constrain non-stomatal sinks with**

**this method. Further research is required to more directly verify whether better constrained $g_s$ leads to improved $v_d$ simulation.**

*554-6: the authors could do a better job at illustrating why they are linking these two ideas*

Response: We clarified our statement as follow:

**The predicted IAV from all four models is smaller than what long-term observations suggest, but its potential contribution to IAV in $O_3$ is still comparable to the long-term variability of background ozone over similar timescales in U.S. summer (Brown-Steiner et al., 2018; Fiore et al., 2014).**

*561-3: yet the authors barely make use of long-term datasets that are available!*

Response: Our intention was to draw attention to the fact that our experiment shows many interesting and notable impacts occurring in parts of the world where there are no available long-term observations to our knowledge, and therefore are these effects are difficult to evaluate. We have clarified by replacing this sentence in question with the following:

** While our results show notable impacts across the globe, in many regions there are no available long-term observation to evaluate the model predictions over interannual timescales.**

*583: what does low baseline vd actually mean?*

Response: Here we were referring to the mean $v_d$ in the unperturbed GEOS-Chem simulation. We deleted the word "baseline" to avoid confusion.

586: v_d

Response: We have made this correction.

*587: do the authors mean the simulation year for the 30% testing?*

Response: We refer to the whole sensitivity simulation. This has been clarified:

**…and possibly the choice of simulation year for the sensitivity simulation…**

*588: is this somewhat inherently in the LAI product?*

Response: This is an interesting question and can be answered in two dimensions. First, LAI retrieval is land cover-dependent (Fang et al., 2013). LAI products spanning before MODIS era

(2000) mostly use static land cover (e.g. Liu et al., 2012; Zhu et al., 2013) that may not even correctly capture the impact land use and land cover change (LULCC) on LAI. Also, at least in those parameterizations, changes in land type causes changes in LAI-independent parameters (e.g. in-canopy aerodynamic resistance, cuticular resistance), which also cannot be captured by LAI changes. In response to the reviewer comment, we have made the following modification to the text:

**…source of variability for $v_d$, and even long-term LAI retrieval (Fang et al., 2013).**

*593-600: as is, this seems like a stretch to me*

Response: We agree that this is a speculative element of our discussion. We meant to emphasize that uncertainty in gaseous dry deposition is not exclusive to $O_3$. We want to encourage similar research attention on the uncertainty in dry deposition of other gases (e.g. $NO_2$, $SO_2$). We have reworded this statement to avoid speculation:

**The impact of dry deposition parameterization choice**  **may also have impacts which we have not explored in this study on other trace gases**….

608: what is the difference between a model-observation integration and an empirical study?

Response: We have reworded this sentence to avoid confusion:

~~This makes a strong case for additional measurements (e.g. Kammer et al., 2019; Li et al., 2018; Stella et al., 2011a), empirical studies (e.g. Ducker et al., 609 2018) and model-observation integrations (e.g. Silva et al., 2019) of ozone dry deposition at different timescales, which would 610 be greatly facilitated by an open data sharing infrastructure (e.g. Baldocchi et al., 2001; Junninen et al., 2009).~~ **This makes a strong case for additional measurement and model studies of ozone dry deposition across different timescales, which would be greatly facilitated by an open data sharing infrastructure (e.g. Baldocchi et al., 2001; Junninen et al., 2009).**

---

## Author Comment (AC2) · 18 Sep 2019

We thank the referee for their positive and constructive comments on our manuscript. We provide our response to each individual reviewer comment (shown in italics) below, including detailed changes to the manuscript (additions in red).

*My only general criticism is that the figures need to be presented in a larger form that will be easier for readers to see.*

Response: We acknowledge that some of the figures are difficult to see, and our manuscript would benefit from addressing this. We have made improvements to Figure 1, 8, and 9 (see below) that we hope will help with readability.

*Line 660 – 661: The blue dots are very difficult to see on these figures. The figures should be made larger!*

Figure 1 has been changed to show larger red dots that make them easier to see.

Furthermore, Figure 8 and Figure 9 have been adjusted to remove white space to allow for larger panels, and zooms into the Earth's land surface by removing areas around the edges where results were minor.

New figure 1:

[Figure]

New figure 8:

[Figure]

New figure 9:

[Figure]

Specific comments:

*p. 1, line 27: Should be "The trend in July ...", not "trends".*

> Response: We have made this correction.

*p. 2, line 62: Should be "... compiled ...".*

> Response: We have made this correction.

*p. 2, line 63: Should be "measurements from the EC and GM ...".*

> Response: We have made this correction.

*p. 7, line 205: Should be "... simulation described in the next sub-section."*

> Response: We have made this correction.

*p. 7, line 211: Should be "... to investigate how ...".*

> Response: We have made this correction.

*p. 7, line 216: Should be "... the increase in atmospheric ...".*

> Response: We have made this correction.

*p. 8, line 233: Should be "... developed by NOAA's National Centers for Environmental Prediction (NCEP) and the NASA Global ...".*

> Response: We changed the sentence to:
>
> **…developed by National Centers for Environmental Prediction (NCEP) of National Oceanic and Atmospheric Administration (NOAA) and…**

*p. 13, line 409: Should be "... We use the Theil-Sen method ...".*

> Response: We have made this correction.

*p. 14, lines 430-431: I believe it should be "... a concomitant decrease in July mean surface ozone ...".*

> Response: We changed the sentence to:
>
> **…a concomitant  decrease in July mean surface ozone…**

*p. 15, line 461: Should be "… as they allow for …".*

Response: We have made this correction.

*p. 16, line 497: Should be "… This suggests that the IAV …".*

Response: We have made this correction.

*p. 17, line 527: Should be "… To better constrain regional dry …".*

Response: We have made this correction.

*p. 17, line 530: Should be "… would be greatly benefited if results of ozone flux measurements were reported as both …".*

Response: In response to comment from another referee, we have already the sentence to:

 **Evaluation and development of ozone dry deposition parameterizations will continue to benefit from publicly available ozone flux measurements and related micrometeorological variables that allow for partitioning measured flux into individual deposition pathways (e.g. Clifton et al., 2017; 2019, Fares et al., 2010, Wu et al., 2018).**

*p. 17, line 538: Should be "… a vast majority of land in …".*

Response: We have made this correction.

*p. 18, line 558: Should be "… mainly concentrates in the drier part of …".*

Response: We have made this correction.

*p. 18, line 562: Should be "… deposition measurements poses …".*

Response: In response to comment from another referee, we have already changed line 562 to:

 **While our results show notable impacts across the globe, in many regions there are no available long-term observation to evaluate the model predictions over interannual timescales.**

*p. 18, line 566: Should be "… The magnitudes of trends are …".*

Response: We have made this correction.

*p. 19, line 597: I believe it should be something like … "… contribute to understanding the role of gaseous dry deposition on air quality, but also to biogeochemical cycling."*

Response: We changed the sentence to:

**…contribute to understanding the  role of gaseous dry deposition  on air quality, but also to biogeochemical  cycling…**

*p. 19, line 600: Should be "… global nitrogen cycles."*

Response: We have made this correction.

*p. 19, line 607: Should be "… scarcity of measurements."*

Response: In response to comment from another referee, we have already changed the sentence to:

** While our results show notable impacts across the globe, in many regions there are no available long-term observation to evaluate the model predictions over interannual timescales.**

---

## Author Response (AR2)

October 15, 2019

RE: Submission acp-2019-429

Dear Dr. Butler (Handling Editor),

Thanks for your positive comments. We are grateful for the opportunity to make these final clarifications in our manuscript. Our response to your notes are as follows:

*Editor Comment:*

*Firstly, in your response to the comment of referee #1 "195: presumably the authors' decisions about land type mapping (& differences for "W89" vs "Z03") impact the authors' results... one implication of this is that the authors' statement in the abstract or introduction that the only thing different across parameterizations is the model structure is not necessarily true", you provide an adequate response to the comment in the main body of your text, but on line 16 of your revised manuscript, it still reads "the differences in simulated vd are entirely due to differences in deposition model structures." It appears to me that differences in the land type mapping also affect the model intercomparison results, so your statement in the abstract may need to be modified to reflect this.*

Author Response:

We appreciate you catching this oversight! We have modified the language in our abstract to:

"We model ozone dry deposition velocities over 1982-2011 using four ozone dry deposition parameterizations that are representative of current approaches in global ozone dry deposition modelling. We use consistent assimilated meteorology, land cover, and satellite-derived leaf area index (LAI) across all four, such that the differences in simulated $v_d$ are entirely due to differences in deposition model structures or assumptions about how land types are treated in each."

*Editor Comment:*

*Secondly, also in the abstract, the text "current approaches of global ozone dry deposition modelling over 1982-2011" suggests that you are using deposition schemes from this period, when I think you actually mean meteorology and LAI from this period. A simple rewording could clarify this.*

Author Response:

We believe his has now been clarified in the text of the abstract as written above.

Moreover, we took the opportunity to correct the following typos (line numbers refer to the tracked-changes version of the manuscript below):

- Line 26: Added an "s" to "tropical rainforests"
- Line 104: Added the work "confirm" to "experiments also confirm CO2 fertilization"
- Line 237: Deleted "which would be an indication of dormant biosphere". This was meant to be removed in response to a reviewer comment about speculation.
- Line 241: In response to the editor's comment above, we have clarified this sentence to read: "so that discrepancies (in space and time) among the predicted vd are dominated by the choice of dry deposition parameterization choice, or assumptions about how land cover is treated."
- Line 399: Removed an extra comma
- Line 531: Added an "s" to "regions"
- Tables 1-3 in the Supplemental Material have been relabeled as "Table S1, S2, S2" instead of "Table A1, A2, A3".

We very much look forward to the publication of our manuscript and appreciate your time. Tracked Changes versions of our manuscript and supplemental material follow this letter.

All the best,

Jeffrey Geddes (Corresponding Author)
Assistant Professor
Department of Earth & Environment
Boston University
jgeddes@bu.edu

[revised manuscript text omitted]

**Supplemental Material**

**Contents:**

**1. Mathematical analysis for sensitivity of $O_3$ to $\Delta v_d / v_d$:**

Assume that $\Delta O_3$ is due to changes in dry deposition flux (with proportionality constant $k_d$) and other first-order processes (e.g. NO titration, loss to $HO_2$ and OH, having total reaction rate $k_c$):

$$dO_3 = d(-k_c O_3 - k_d v_d O_3) \ (S1)$$

Here, $k_c$ and $k_d$ (which are related to meteorology and concentration of other relevant chemical species), are assumed to be relatively constant, so that that the perturbation in $v_d$ does not trigger significant non-linearity. Expanding the differential and rearranging the terms yields:

$$\frac{dO_3}{O_3} = \frac{-k_d \ dv_d}{1 + k_c + k_d} \ (S2)$$

Integrating S2 between perturbed $(O_3 + \Delta O_3, v + \Delta v_d)$ and unperturbed $(O_3$ and $v_d)$ values yields:

$$\ln\left(1 + \frac{\Delta O_3}{O_3}\right) = -\ln\left(1 + \frac{k_d \Delta v_d}{1 + k_c + k_d v_d}\right) \ (S3)$$

Since $\Delta O_3$ is small compared to $O_{3,0}$, first-order expansion is valid. When $\Delta v_d$ is small enough relative to $v_d$ for first-order approximation, Taylor's expansion of S4 yield:

$$\frac{\Delta O_3}{O_3} = -\frac{k_d}{1 + k_c + k_d v_d} \Delta v_d \ (S4)$$

S5 can be rearranged to yield:

$$\Delta O_3 = -\frac{k_d v_d O_3}{1 + k_c + k_d v_d} \frac{\Delta v_d}{v_d} = \beta \frac{\Delta v_d}{v_d}, where \ \beta = -\frac{k_d v_d O_3}{1 + k_c + k_d v_d} < 0 \ (S5)$$

This shows that when the $\Delta v_d / v_d$ is small enough $(\ln(1+x) \approx x)$ and does not cause non-linearity ($k_c$ and $k_d$ = constant) in chemistry, $\Delta O_3$ is linearly proportional to $\Delta v_d / v_d$. The error of linearizing the natural logarithms equals to the difference between $\ln(1+x)$ and $x$. This analysis gives the conditions for when the first-order approximation is reasonable, and allowing us to estimate the error when deviating from these condition. Assuming $\beta$ is correctly estimated by chemical transport model, the error of linearization at $\Delta v_d / v_d = \pm 50\%$ (the upper bound of $\Delta v_d / v_d$ consistent with our analysis), is on the order of 25%. For more typical value of $\Delta v_d / v_d$ (20%), the error is around 10%.

As $\Delta v_d / v_d$ gets larger, we can expand R.H.S of S3 to the second order and investigate sensitivity of $\Delta O_3$ to $\Delta v_d / v_d$:

$$\Delta O_3 = \beta \frac{\Delta v_d}{v_d} - \frac{\beta^2}{2 O_3}\left(\frac{\Delta v_d}{v_d}\right)^2 = \left(\beta - \frac{\beta^2}{2 O_3} \frac{\Delta v_d}{v_d}\right)\left(\frac{\Delta v_d}{v_d}\right) = \beta' \frac{\Delta v_d}{v_d} \ (S6)$$

Where $\beta'$ is the "corrected $\beta$", which is a function of $\Delta v_d / v_d$.

To illustrate the potential impact of such non-linearity on $\Delta O_3$, we compare July $\Delta O_{3,Z03\_BB}$ estimated using first-order estimation with $\beta$ derived from $\Delta v_d / v_d = +15\%$ (fig. S1b) and $+30\%$ (fig. S1a), and second-order approximation (fig. S1c), and the result is shown in figure S1. The three different methods produce very similar $\Delta O_3$, and their differences have little impact on our conclusion. For simplicity, we only show the result using $\beta$ derived from $\Delta v_d / v_d = +30\%$ in the main manuscript.

As noted above and in the main manuscript, our approach is limited by the assumption that chemistry and transport do not introduce non-linear terms which may not be realistic. Rather, our approach is intended to identify hotspots of impact, and quantify these potential impacts to a first order. More rigorous modeling efforts could then be targeted in future work.

[Figure]

**Figure S1. July $\Delta O_{3,Z03\_BB}$ calculated using a) first-order method where $\beta$ is derived from $\Delta v_d/v_d = +30\%$ GC sensitivity run, b) first order method where $\beta$ is derived from $\Delta v_d/v_d = +15\%$ GC sensitivity run, and c) second-order method with $\beta$ derived from $\Delta v_d/v_d = +15\%$.**

**2. A brief description of photosynthesis-stomatal conductance ($A_n$-$g_s$) module in TEMIR (a manuscript is in prep)**

TEMIR largely follows Oleson et al. (2013), where net photosynthetic rate ($A_n$, µmol $CO_2$ m$^{-2}$ s$^{-1}$), stomatal conductance for water ($g_{sw}$, µmol m$^{-2}$ s$^{-1}$) and $CO_2$ concentration in leaf mesophyll ($c_i$, mol mol$^{-1}$) are solved simultaneously by the following coupled set of equations:

$$A_n = \frac{g_{sw}}{1.6}(c_a - c_i) \; (S7)$$

$$g_{sw} = \beta_t g_0 + g_1 \frac{A_n}{c_s} RH_s \; (S8)$$

$$A_n = A - R_d \; (S9)$$

Here, $c_a$ is $CO_2$ concentration (mol mol$^{-1}$), $\beta_t$ is soil moisture stress factor (unitless), $g_0$ is minimum stomatal conductance (µmol m$^{-2}$ s$^{-1}$), $A_n$ is net photosynthetic rate (µmol $CO_2$ m$^{-2}$ s$^{-1}$), $A$ is gross photosynthetic rate (µmol $CO_2$ m$^{-2}$ s$^{-1}$) and $R_d$ is dark respiration rate (µmol $CO_2$ m$^{-2}$ s$^{-1}$). Furthermore, $c_s$ and $RH_s$ are the $CO_2$ concentration (mol mol$^{-1}$) and relative humidity (unitless) at leaf surface. $A$ is calculated following Bonan et al. (2011), which is based on Farquhar et al. (1980) and Collatz et al. (1992):

$$\Theta_{cj}A_i^2 - (A_c + A_j)A_i + A_c A_j = 0 \; (S10)$$

$$\Theta_{ip}A^2 - (A_i + A_p)A + A_i A_p = 0 \; (S11)$$

For C3 plants, $\Theta_{cj} = 0.98$ and $\Theta_{ip} = 0.95$. For C4 plants, $\Theta_{cj} = 0.80$ and $\Theta_{ip} = 0.95$. Rubisco-limited rate ($A_c$, µmol $CO_2$ m$^{-2}$ s$^{-1}$), light-limited rate ($A_j$, µmol $CO_2$ m$^{-2}$ s$^{-1}$), product-limited rate ($A_p$, µmol $CO_2$ m$^{-2}$ s$^{-1}$) and $R_d$ are calculated as:

$$A_c = \begin{cases} \dfrac{V_{c\,max}(c_i - \Gamma_*)}{c_i + K_c(1 + \dfrac{0.21P_{atm}}{K_o})} & for\ C_3\ plants \\ V_{c\,max} & for\ C_4\ plants \end{cases} \quad (S12)$$

$$A_j = \begin{cases} \dfrac{J(c_i - \Gamma_*)}{4c_i + 8\Gamma_*} & for\ C_3\ plants \\ 0.23\phi & for\ C_4\ plants \end{cases} \quad (S13)$$

$$A_c = \begin{cases} 3T_p & for\ C_3\ plants \\ k_p \dfrac{c_i}{P_{atm}} & for\ C_4\ plants \end{cases} \quad (S14)$$

$$R_d = \begin{cases} 0.015V_{c\,max} & for\ C_3\ plants \\ 0.025V_{c\,max} & for\ C_4\ plants \end{cases} \quad (S15)$$

Here, $V_{cmax}$, $\Gamma_*$, $P_{atm}$, $J$, $\varphi$, $T_p$ and $k_p$ are the maximum rate of carboxylation (µmol m⁻² s⁻¹), CO₂ compensation point (mol mol⁻¹), atmospheric pressure (Pa), electron transport rate (µmol m⁻² s⁻¹), absorbed photosynthetically active radiation (PAR) (W m⁻²), triose phosphate utilization rate (µmol m⁻² s⁻¹) and initial slope of C₄ CO₂ response curve (µmol Pa⁻¹ m⁻² s⁻¹). $K_c$ and $K_o$ are the Michaelis-Menten constants for CO₂ and O₂ (Pa). Furthermore, $J$ is calculated as the smaller root of the following equation:

$$0.7J^2 + (1.955\phi + J_{max})J + 1.955\phi = 0 \ (S16)$$

Where $J_{max}$ is the maximum potential rate of electron transport (µmol m⁻² s⁻¹). As $J_{max}$, $\varphi$, $V_{cmax}$ and the variables related to $V_{cmax}$ ($\Gamma_*$, $J_{max}$, $T_p$, $R_d$) differ between sunlit and shaded leaves, the above set of equations are solved separately for sunlit and shaded leaves.

The parameters ($V_{cmax}$, $\Gamma_*$, $K_c$, $K_o$, $J_{max}$, $T_p$, $R_d$) are functions of vegetation temperature ($T_v$), and the temperature scaling formulae are given at eq. 8.9 to eq. 8.14, while the effect of temperature acclimation (Kattge and Knorr, 2007) on $J_{max}$ and $V_{cmax}$ are given at eq. 8.15 and 8.16 in Oleson et al. (2013). Other details of the model formalism (e.g. canopy scaling and effect of $\beta_t$ on $V_{cmax}$) also follow Chapter 8 in Oleson et al. (2013), therefore we will focus on describing the main differences between CLM 4.5 and TEMIR.

First, TEMIR is driven entirely by assimilated meteorology. Instead of solving the whole surface energy balance equation, TEMIR consistently calculates $T_v$ from 2-meter air temperature ($T_2$, K) and sensible heat flux ($H$, W m⁻²) using Monin-Obukhov similarity theory (Monin and Obukhov, 1954):

$$T_v = T_2 + \frac{H}{\rho c_p}(r_{a,h} + r_{b,h}) \ (S16)$$

Where $\rho$, $c_p$, $r_{a,h}$ and $r_{b,h}$ are air density (kg m⁻³), specific heat of air at constant pressure (J kg⁻¹ K⁻¹), aerodynamic and laminar boundary-layer resistance (s m⁻¹) of heat, respectively.

Secondly, MERRA-2 only provides soil moisture output at two levels (surface and root zone), which is not compatible with the multi-layer soil module in CLM. Therefore, instead of aggregating $\beta_t$ from multiple soil layers, TEMIR calculates $\beta_t$ from the root-zone soil wetness of MERRA-2. Soil wetness ($s$) is first converted into soil matric potential ($\psi$, mm) using the following equation:

$$\psi = \psi_{sat}s^{-B} \ (S17)$$

Where $\psi_{sat}$ and $B$ are the soil matric potential (mm) at saturation and Clapp-Hornberger exponent (Clapp and Hornberger, 1978), which are related to soil property. Then $\beta_t$ is calculated as:

$$\beta_t = \frac{\psi_c - \psi}{\psi_c - \psi_0} \left( \frac{\theta_{sat} - \theta_{ice}}{\theta_{sat}} \right), 0 \leq \beta_t \leq 1 \ (S18)$$

Where $\psi_c$ and $\psi_0$ are the soil matric potential (mm) at which stomata are full close or fully open, and the term in the bracket account for the fact that frozen water are not available for plants.

[Figure]

**Figure S2. July average soil moisture stress factor ($\beta_t$). $\beta_t = 1$ represents no soil moisture stress, while smaller $\beta_t$ means stronger soil moisture stress and more stomatal closure. $\beta_t = 0$ signifies that soil moisture stress is so strong that it completely shuts down stomatal activity.**

**3. Table A1 to Table A3**

| | W98 | Z03 | W98_BB | Z03_BB |
|---|---|---|---|---|
| $R_a$ | $R_a = \frac{1}{\kappa u_*}\left[\ln(\frac{z}{z_0}) - \Psi\left(\frac{z}{L}\right) + \Psi(\frac{z_0}{L})\right]$
 When $\varsigma \geq 0, \Psi(\varsigma) = -5\varsigma$
 When $\varsigma < 0, \Psi(\varsigma) = 2\ln(\frac{1+\sqrt{1-16\varsigma}}{2})$ | | | |
| $R_b$ | $R_b = \frac{2}{\kappa u_*}(\frac{Sc}{Pr})^{2/3}$ | | | |
| $R_s$ | $R_s = r_s(PAR, LAI)f_T \frac{D_{H_2O}}{D_{O_3}}$ | $R_s = \frac{r_s(PAR,LAI)}{(1-w_{st})f_T f_{vpd} f_\psi} \frac{D_{H_2O}}{D_{O_3}}$ | $g_s = g_0 + m\frac{A_n}{C_s}h_s$
 $R_s = \frac{1}{g_s}\frac{D_{H_2O}}{D_{O_3}}$ | $g_s = g_0 + m\frac{A_n}{C_s}h_s$
 $R_s = \frac{1}{(1-w_{st})g_s}\frac{D_{H_2O}}{D_{O_3}}$ |
| Cuticular Resistance ($R_{cut}$) | $R_{cut} = \frac{R_{cut_0}}{LAI}$ | For dry surface,
 $R_{cut} = \frac{R_{cut_{d0}}}{e^{0.03RH}LAI^{0.25}u_*}$
 For wet surface,
 $R_{cut} = \frac{R_{cut_{w0}}}{LAI^{0.5}u_*}$ | Same as W98 | Same as Z03 |
| In-canopy aerodynamic resistance ($R_{ac}$) | Prescribed | $R_{ac} = R_{ac_0}\frac{LAI^{0.25}}{u_*}$ | | |
| Ground Resistance ($R_g$) | Prescribed | | | |
| Lower-canopy aerodynamic resistance ($R_{alc}$) | $R_{alc} = 100(1 + \frac{1000}{R+10})$ | - | | |
| Lower-canopy surface resistance ($R_{clc}$) | Prescribed | - | | |

[revised manuscript text omitted]

---

## Author Response (AR3)

October 22, 2019

RE: Submission acp-2019-429

Dear Dr. Butler (Handling Editor),

We are thrilled that our manuscript has been accepted for publication. Thanks for your time and effort during the submission process.

We hope that it is acceptable to you that we have made one final minor change to the manuscript in our finalized version. To acknowledge the recent publication of a relevant article that appeared in the September 2019 issue of *Journal of Geophysical Research Atmospheres*, we have added a citation to this work in the conclusion section of our study. In the finalized manuscript, lines 628-635 now read:

> "We demonstrate that the parameterizations with explicit dependence on hydroclimatic variables have higher sensitivity to climate variability than those without. **Lin et al. (2019) likewise recently demonstrated the importance of accounting for water availability in $O_3$ dry deposition modeling.** Difficulties in evaluating predictions…"

We are uploading our finalized manuscript, supplement, and figures. We look forward to seeing the proofs once they are available.

Thanks again,

Jeffrey Geddes (Corresponding Author)
Assistant Professor
Department of Earth & Environment
Boston University
jgeddes@bu.edu